# Immune-induced TCR-like antibodies regulate specific T cell response in mice

Kazuki Kishida[1], Keisuke Kawakami [2], Hiroaki Tanabe[3,4], Wataru Nakai[1,9], Koji Yonekura [2,5], Shigeyuki Yokoyama[3,4] & Hisashi Arase [1,6,7,8] ✉

Antigen-specific regulation of T cell response is crucial for limiting hyper-immune response. However, the molecular mechanisms governing specific immune regulation remain unclear. In this study, we discover that antibodies specific to the antigen peptide-MHC class II complex are produced during helper T cell responses to various antigens, including hen egg lysozyme and proteolipid protein peptide. These antibodies specifically inhibit T cell receptor (TCR) recognition of MHC class II molecules presenting specific antigen peptide. We term these antibodies 'immune-induced TCR-like anti-bodies' or iTabs. Immunization with peptides containing flanking residues induces iTabs whereas immunization with peptides lacking flanking residues does not. Furthermore, we show that immunization with iTab-inducible pep-tide or iTab treatment suppress autoimmune disease development in a mouse model of experimental autoimmune encephalomyelitis. Thus, our findings provide a strategy for suppressing antigen-specific helper T cell responses using specific peptides, potentially controlling autoimmune diseases.

Immunization with antigens triggers antigen-specific T cell and B cell activation, leading to antibody production against the antigens. Major histocompatibility complex class II molecules (MHC-II) are crucial for adaptive immunity, presenting peptide antigens to helper T cells. Certain MHC-II alleles are strongly associated with the risk of auto-immune diseases and allergies[1,2]. Thus, controlling T cell receptor (TCR) recognition of peptides presented on MHC-II is vital for con-trolling the immune response[3,4].

Antibodies that recognize the peptide-MHC-II complex, exhibit-ing similar specificity to TCRs, can be generated by immunizing with the peptide-MHC-II complex, providing a tool for analyzing antigen presentation[5,6] and blocking TCR recognition[7,8]. However, it has been generally thought that these TCR-like antibodies are not typically produced during normal immune responses.

The peptide-binding groove of MHC-II is open at both ends, resulting in fewer restrictions on the peptide length[9]. Indeed, not only peptides but also whole proteins can be presented on MHC-II[10-13]. Although minimal peptides suffice for helper T cell recognition and are commonly used in studies, naturally presented peptides on MHC-II often possess long flanking residues (FR), whose roles in immunity are not fully understood[14-16].

In this study, we examine how peptide flanking residues influence the induction of TCR-like antibodies during antigen-specific immune responses and assess whether these antibodies modulate CD4+ T cell

[1]Department of Immunochemistry, Research Institute for Microbial Diseases, The University of Osaka, Suita, Osaka, Japan. [2]Biostructural Mechanism Group, RIKEN SPring-8 Center, Hyogo, Japan. [3]Department of Drug Target Protein Research, Shinshu University School of Medicine, Matsumoto, Nagano, Japan. [4]Department of Structural Biology and Biochemistry, Institute of New Industry Incubation, Institute of Science Tokyo, Tokyo, Japan. [5]Institute of Multi-disciplinary Research for Advanced Materials, Tohoku University, 2-1-1 Katahira, Aoba-ku, Sendai, Miyagi, Japan. [6]Laboratory of Immunochemistry, WPI Immunology Frontier Research Center, The University of Osaka, Suita, Osaka, Japan. [7]Center for Advanced Modalities and DDS, The University of Osaka, Suita, Osaka, Japan. [8]Center for Infectious Disease Education and Research, The University of Osaka, Suita, Osaka, Japan. [9]Present address: Laboratory for Innate Immune Systems, Department of Microbiology and Immunology, Graduate School of Medicine, The University of Osaka, Suita, Osaka, Japan. ✉e-mail: arase@biken.osaka-u.ac.jp

activation. By comparing immune responses elicited by peptides with or without flanking residues, we demonstrate that flanking residues critically shape the generation, specificity, and functional capacity of TCR-like antibodies. We further show that antibodies induced in the presence of flanking residues more effectively recognize peptide–MHC-II complexes and alter antigen-specific CD4⁺ T cell responses. These findings reveal an unappreciated role for peptide flanking residues in shaping humoral recognition of peptide–MHC-II complexes. Our results provide conceptual and practical insights for the design of immunogens aimed at selectively enhancing or regulating antigen-specific CD4⁺ T cell responses.

## Results

### Immunization with protein antigen induces antibodies against the peptide-MHC-II complex

We examined whether a general antigen-specific immune response produces antibodies against the peptide-MHC-II complex. B10.A mice (I-A$^k$) were immunized with hen egg lysozyme (HEL) protein. We then analyzed serum antibody binding to 30-amino-acid overlapping HEL peptides pulsed onto 293 T cells expressing MHC-II, I-A$^k$. Serum antibodies recognized cells pulsed with HEL$_{41-70}$ and HEL$_{51-80}$ peptides, both of which contain the major T cell epitope, HEL$_{48-61}$[17] (Fig. 1a). In contrast, the antibodies did not recognize cells pulsed with HEL$_{48-61}$, which contains only the major T cell epitope. Notably, HEL$_{48-61}$ was efficiently presented by MHC-II, as confirmed by binding of the monoclonal antibody (mAb) Aw3.18, which specifically recognizes HEL peptide–MHC-II complexes[5] (Fig. 1b).

Similar to the Aw3.18 mAb, a single-chain TCR-Fc fusion protein of the 3A9 TCR, specific to HEL$_{48-61}$ peptides presented on MHC-II (3A9 TCR-Fc)[18], also bound to HEL$_{48-61}$ peptide-pulsed cells in a dose-dependent manner (Supplementary Fig. 1a and Fig. 1b). When HEL$_{48-61}$ peptides with additional N-terminal (HEL$_{41-61}$) or C-terminal (HEL$_{48-70}$) FR were pulsed, the serum antibodies bound to HEL$_{48-70}$ with a C-terminal, but not N-terminal FR. At least two tryptophan residues at the C-terminal FR are necessary for the antibody binding (Fig. 1c). Collectively, these data indicate that immunization with HEL protein induces the antibodies that recognize HEL peptides containing the C-terminal FR presented on MHC-II expressing cells.

Consistent with these findings, antibodies recognizing the chicken ovalbumin (OVA) peptide–I-A$^d$ complex were detected in sera from BALB/c mice immunized with OVA protein (Supplementary Fig. 2a), and antibodies recognizing the myelin oligodendrocyte glycoprotein (MOG) peptide–I-A$^b$ complex were detected in sera from C57BL/6 mice immunized with MOG protein (Supplementary Fig. 2b). In contrast, antibodies recognizing peptide-MHC-II complex were not detected in sera from non-immunized mice. Together, these data suggest such antibodies can be induced by immunization with commonly used protein antigens across multiple mouse strains and MHC class II alleles.

Next, we investigated whether the antibodies produced by HEL protein immunization recognize the FR of HEL peptides alone or the entire peptide-MHC-II complex. HEL protein-immunized serum antibodies were absorbed on entire HEL protein- or HEL$_{48-64}$ peptide-bound Sepharose beads (Fig. 1d). Upon absorption, the antibody titers against the HEL protein or the HEL$_{48-64}$ peptide itself, but not against HEL peptides presented on MHC-II, were decreased (Fig. 1d). This indicated that antibodies specific to the peptide-MHC-II complex were generated by HEL protein immunization. Because these antibodies recognize both the peptide and MHC-II, similar to TCRs, we termed them immune-induced TCR-like antibody (iTab).

Interestingly, iTab titers were decreased from two weeks post-immunization, whereas anti-HEL antibody titers simultaneously increased (Fig. 1e). This suggests that the dynamics of iTab production differ from those of a general antibody response. To determine whether iTab production depends on CD4⁺ T cell help during the HEL response, we depleted CD4⁺ T cells prior to HEL immunization using an anti-CD4 antibody. iTab production was completely abolished in CD4⁺ T cell–depleted mice (Supplementary Fig. 2c), indicating that T cell help is required for iTab induction, as is typical for conventional IgG responses. Moreover, iTabs were undetectable in non-immunized serum, and sequence analysis of iTabs revealed somatic mutations in variable regions of both heavy and light chains, suggesting that iTabs undergo T cell-dependent affinity maturation.

Since both TCRs and iTab bind to peptides presented on MHC-II, we investigated whether iTabs block TCR recognition. 3A9 TCR-Fc binding to HEL$_{48-64}$-loaded cells was blocked dose-dependently by iTabs from HEL protein-immunized mice (Fig. 1f). However, the binding of TCR-Fc to HEL$_{48-61}$-loaded cells was not blocked by the iTab. This indicated that iTabs that block anti-HEL T cell recognition were produced during the immune response to HEL protein. We performed LC–MS/MS analysis of MHC-II–bound peptides and found that HEL-derived peptides containing C-terminal flanking residues, including HEL$_{48-62}$ and HEL$_{48-63}$, are frequently presented by MHC-II, consistent with a previous report[17]. (Supplementary Fig. 2d and Supplementary Table 1). This result indicates that, within the naturally processed peptide pool, flanking-residue–containing peptides are present and therefore provide physiologically relevant targets for iTab recognition. Immunization of B10 (I-A$^b$), B10.D2 (I-A$^d$), and B10.S (I-A$^s$) congenic mice with HEL protein induced antibodies against the HEL protein itself. However, antibodies in sera from these immunized mice did not bind to HEL$_{48-64}$ pulsed cells and therefore did not block 3A9 TCR-Fc binding (Fig. 1g). These data indicate that the iTabs exhibit MHC-restricted recognition similar to TCR recognition.

### Peptide FR is required for iTab production

We next examined whether HEL peptide immunization induces iTabs like HEL protein immunization (Supplementary Fig. 3a). Immunization with HEL$_{48-70}$ or HEL$_{48-64}$ (FR⁺ HEL), containing C-terminal FR, but not HEL$_{40-61}$ or HEL$_{48-61}$ (FR⁻ HEL), lacking C-terminal FR, induced iTabs that bind to cells pulsed with HEL$_{48-64}$ (Fig. 2a). iTabs did not bind to cells pulsed with HEL$_{40-61}$ (Fig. 2a). These data indicate that C-terminal FR is essential for iTab induction. Furthermore, immunization with HEL peptides containing substitutions within the MHC-binding groove failed to induce antibodies that recognize the wild-type FR⁺ HEL peptide-MHC-II complex (Fig. 2b). In contrast, immunization with the mutant peptide HEL48-61 (N59W/S60W) induced antibodies specific to the corresponding mutant peptide-MHC-II complex (Supplementary Fig. 3b). Together, these results suggest that iTab induction and specificity are highly dependent on the precise pMHC-II conformation and support a critical role for flanking residues in generating antibodies that recognize the wild-type FR⁺ pMHC-II complex.

To test whether iTab induction requires a C-terminal flanking region or can also be supported by an N-terminal flanking region, we analyzed the *Schistosoma mansoni* epitope Sm-P40$_{234-246}$, which is presented by I-A$^k$ as peptides containing an N-terminal flanking residue[19] (Supplementary Fig. 3c). Immunization with Sm-P40$_{234-246}$ induced iTabs against the Sm-P40$_{234-246}$-MHC-II complex, but not the Sm-P40$_{237-249}$-MHC-II complex (Fig. 2c). On the other hand, immunization with Sm-P40$_{237-246}$ and Sm-P40$_{237-249}$ did not induce iTabs against the Sm-P40-MHC-II complex. This suggests that N-terminal FR is necessary for inducing iTabs, and C-terminal FR is not required, unlike the HEL peptides.

Based on our finding that the C-terminal WWA is required for HEL to induce iTabs, whereas the N-terminal PKS is required for Sm-P40, we generated FR-swapped chimeric peptides (PKS–HEL and Sm-P40–WWA) to test whether the requirement for FRs is transferable across distinct core epitopes (Supplementary Fig. 3c). HEL peptide with N-terminal PKS FR induced antibodies against the Sm-P40$_{234-246}$-MHC-II complex (Fig. 2c). Similarly, Sm-P40 peptide with the C-terminal WWA FR elicited antibodies against the HEL$_{48-64}$-MHC-II

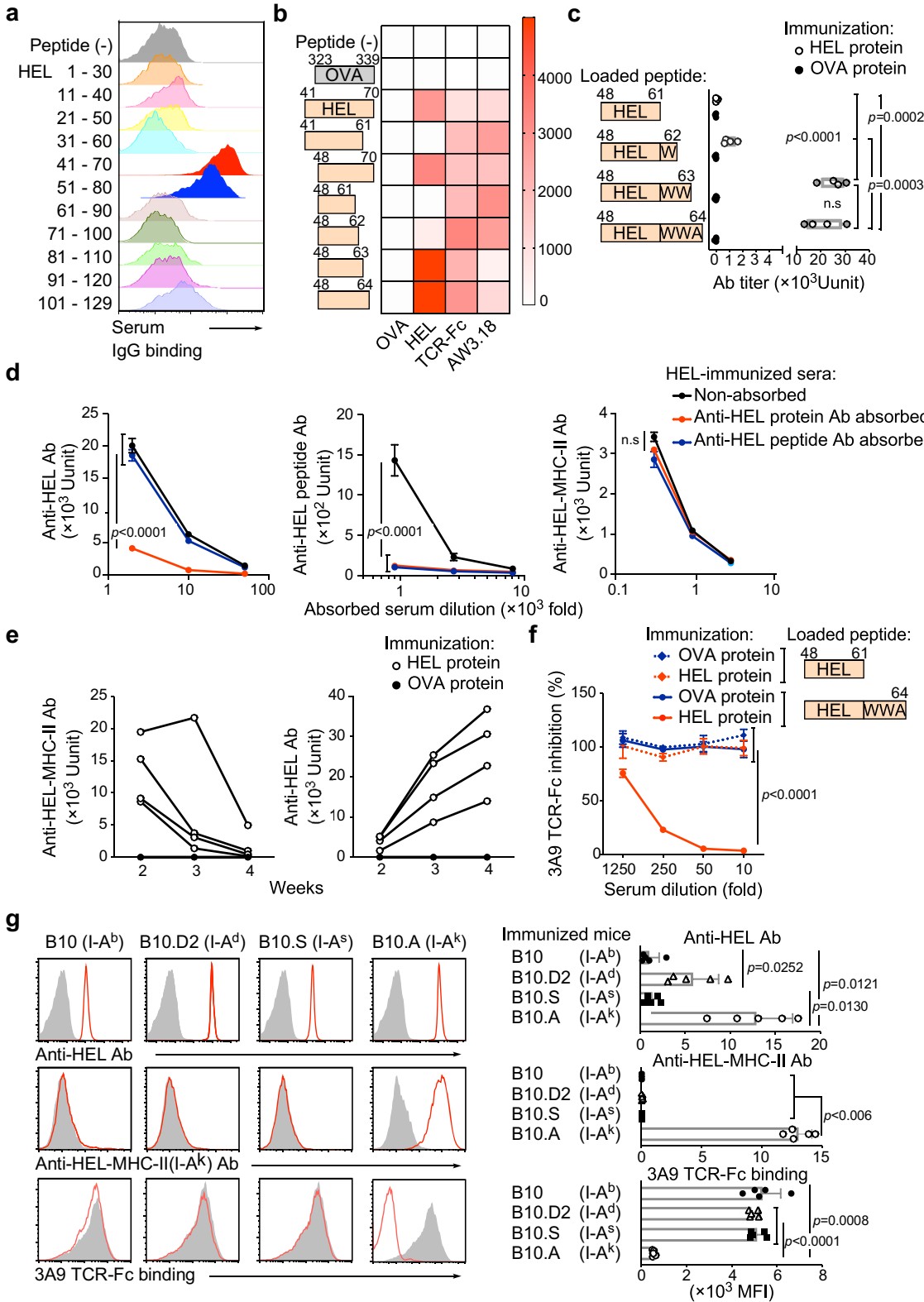

complex (Fig. 2c). These results indicated that the FR is sufficient for iTabs production, although the requirement of N- or C-terminal FR depends on the antigen.

Next, we analyzed the role of the FR on the naturally processed peptides on MHC-II. Invariant chain (Ii) and H2-M are essential molecules for peptide presentation by MHC-II[9]. We generated Ii or H2-Mα knockout LK35.2 cell lines expressing I-A$^k$ (Supplementary Fig. 4a). When wild-type LK35.2 cells were incubated with HEL protein, Aw3.18

antibody and 3A9 TCR-Fc bound to the cells, indicating that naturally processed HEL peptides were presented on MHC-II (Supplementary Fig. 4b). Moreover, serum iTabs from mice immunized with FR$^+$ HEL peptide bound to wild-type LK35.2 cells pulsed with HEL protein, but serum antibodies from mice immunized with FR$^-$ HEL peptide did not (Fig. 3a). In contrast, when Ii or H2-Mα knockout LK35.2 cells were incubated with HEL protein, Aw3.18, 3A9 TCR-Fc and the serum iTabs did not bind to the cells (Supplementary Fig. 4b and Fig. 3a). These

**Fig. 1 | Production of iTabs during immune response to HEL protein. a** The binding of serum antibodies (Abs) from HEL protein-immunized mice to HEL peptide-pulsed MHC-II expressing 293 T cells was analyzed by flow cytometry. **b** Mean fluorescence intensities of antibody (Ab) binding are presented as a heat-map. Serum Abs from OVA-immunized mice, 3A9 TCR-Fc, and Aw3.18 Ab were used as controls. **c** Serum Ab titers against HEL peptide-pulsed MHC-II expressing 293 T cells are shown (n = 4 per group). **d** Antibody titers against HEL protein (left), HEL peptide (center), and HEL48–64 peptide-loaded MHC-II expressing cells (right) (n = 3 technical replicates). **e** Time course of anti-HEL iTab (left) or anti-HEL Ab (right) titers after HEL protein immunization. Serum from OVA-immunized mice

was analyzed as a control (n = 4 per group). **f** Blocking of 3A9 TCR-Fc binding to HEL peptide-loaded MHC-II expressing cells by serum Abs from HEL or OVA protein-immunized mice (n = 3 technical replicates). **g** iTab production in B10 congenic mice. Anti-HEL protein Abs (red line, upper row), anti-HEL iTabs (red line, middle row), and blocking of 3A9 TCR-Fc binding by iTabs (red line, lower row) are shown (n = 5 per group). Non-immunized mice serum was used as controls (shaded histogram). Data are presented as mean values (min−max) (**c**) and mean values +/− SD (**d, f** and **g**). Statistical significance was calculated by one-way ANOVA (**c** and **g**) and two-way ANOVA (**d** and **f**) followed by Tukey's correction. All data are representative of three or more experiments. n.s: no statistical significance.

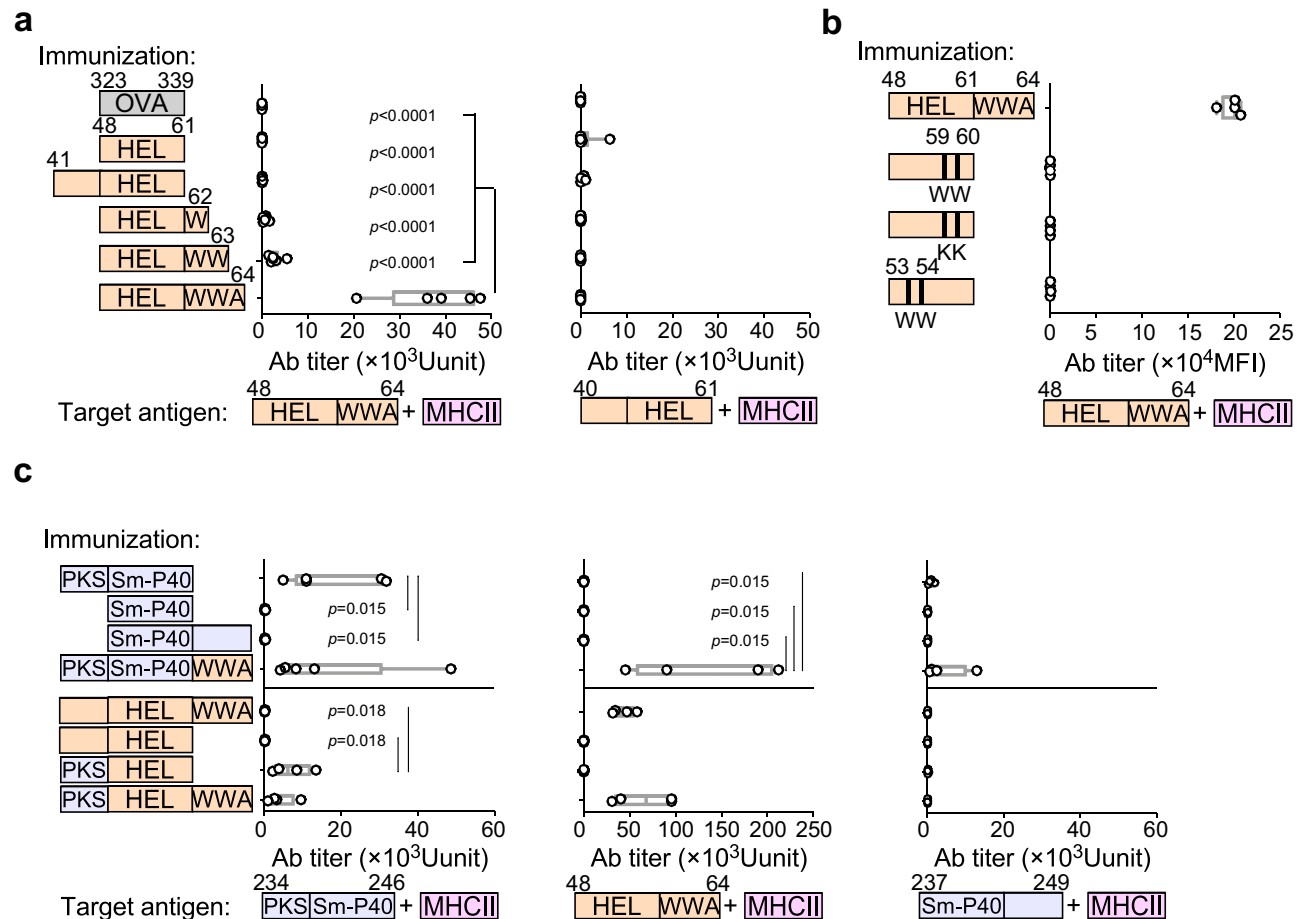

**Fig. 2 | Production of iTabs during immune response to antigen peptides.**
**a** Serum iTab titers induced by immunization with FR+ or FR− HEL peptides. Ab titers at different lengths of peptides presented on MHC-II (n = 4 for OVA323–339 and HEL41–61 immunization group, n = 5 for HEL48–61 and HEL48–64 immunization group, n = 6 for HEL48–62 and HEL48–63 immunization group). **b** Serum iTab titers induced by immunization with mutated FR− HEL peptides (n = 4 mice per group). **c** Serum iTab titers induced by immunization with HEL, Sm-P40 or hybrid peptides. iTab

titers against FR+ Sm-P40 (left), FR+ HEL (middle) and FR− Sm-P40 peptide (right)-pulsed MHC-II expressing cells (n = 5 for SmP40234–246 and SmP40234–246−WWA immunization group, n = 4 for other immunization group). p-values were determined by one-way ANOVA with Tukey's correction (**a**) and Bonferroni's correction (**b**). Data are shown as mean (min−max). Data represent at least three independent experiments.

results indicated that peptides with FR are naturally presented on MHC-II during normal antigen processing in antigen-presenting cells (APC) and are recognized by iTabs.

## iTabs suppress antigen-specific helper T cell response

iTabs induced by HEL protein immunization blocked 3A9 TCR-Fc binding to FR+ HEL peptide presented on MHC-II, suggesting that iTabs can suppress antigen-specific helper T cell responses. iTabs induced by immunization with FR+ HEL peptide but not FR− HEL peptide also blocked 3A9 TCR-Fc binding to HEL protein pulsed APCs (Supplementary Fig. 4c). Furthermore, activation of 3A9 TCR GFP reporter

cells was blocked by serum iTabs induced by FR+ HEL peptide immunization (Fig. 3b, c). Notably, polyclonal IgG purified from sera of mice immunized with FR+ peptide suppressed antigen-specific T cell responses. In contrast, sera from mice immunized with FR− peptide failed to exhibit such suppression, despite containing anti-peptide antibodies. These findings further support the conclusion that anti-peptide antibodies do not play a primary role in mediating T cell suppression. To further analyze the fine specificity of iTabs, we generated a monoclonal iTab. The hybridomas were established from unselected, whole draining lymph nodes from FR+ HEL peptide-immunized mice (Supplementary Fig. 5a). The monoclonal HEL-iTab

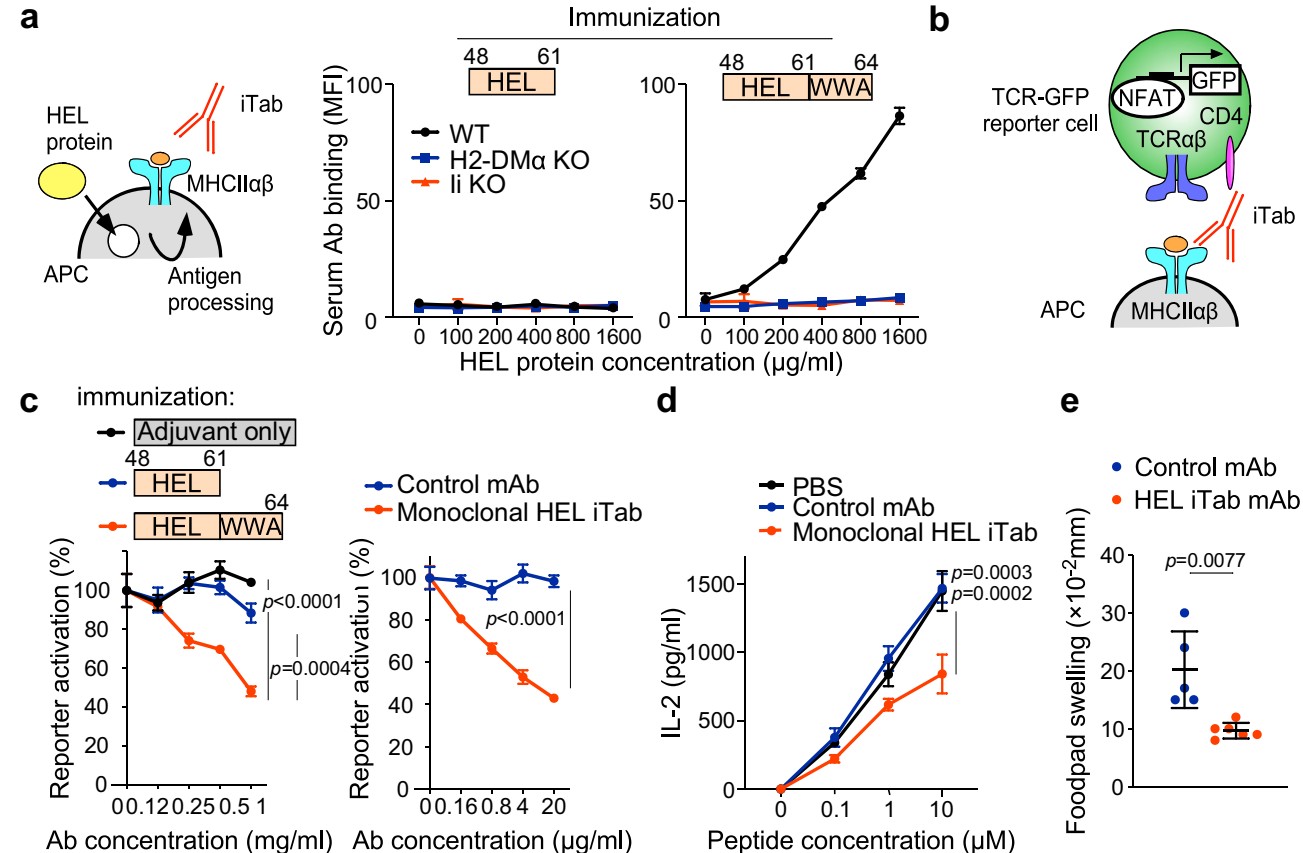

**Fig. 3 | Blocking of antigen-specific T cell response by iTabs. a** Binding of iTabs induced by FR⁻ (left) or FR⁺ (right) HEL peptide immunization was analyzed against wild-type (WT) (black line), H2-DMα (blue line) or invariant chain (red line) knockout cells pulsed with HEL protein. **b** TCR NFAT-GFP reporter system. **c** Inhibition of 3A9 TCR reporter cell activation by iTabs from FR⁻ or FR⁺ HEL peptide- immunized mice (left) and monoclonal anti-HEL iTab 11–72 (right).

**d** Inhibition of IL-2 production by HEL-specific T cells from mice immunized with FR⁺ HEL peptide by anti-HEL iTab (a, c and d, n = 3 technical replicates). **e** Inhibition of DTH response by anti-HEL iTab 11–72 (n = 5 for control group, n = 6 for iTab group). p-values were determined by one-way ANOVA with Tukey's correction (**c**, **d**), except for (**e**), which was analyzed using the Mann-Whitney U test. Bars represent means +/− SD; Data represent at least three independent experiments.

11–72 recognized the HEL peptide-MHC-II complex but not the MHC-II (Supplementary Fig. 5b) or the peptide alone (Supplementary Fig. 5c) and blocked 3A9 TCR reporter cell activation (Fig. 3c). Notably, 11–72 mAb blocked the binding of 3A9 TCR-Fc stronger than Aw3.18 mAb that recognizes HEL peptides presented on MHC-II (Supplementary Fig. 5d). These results suggest that the iTabs are involved in the inhibition of antigen-specific T cell activation.

To analyze the regulatory capacity of iTabs in vivo, FR⁺ HEL peptide-immunized mice were treated with the anti-HEL iTab, 11–72. T cells harvested from anti-HEL iTab-treated mice showed a reduced IL-2 production compared to T cells from control mAb-treated mice (Fig. 3d). Moreover, the delayed-type hypersensitivity (DTH) response, a CD4⁺ T cell-dependent inflammatory reaction, was also reduced in anti-HEL iTab-treated mice (Fig. 3e). These results suggest that iTabs can suppress antigen-specific immune responses in vivo.

We next investigated whether iTabs can mediate Fc-dependent cytotoxicity against antigen-presenting cells. In vitro, APCs pulsed with the FR⁺ HEL peptide exhibited increased killing in an iTab concentration-dependent manner (Supplementary Fig. 5e). For in vivo cytotoxicity assays, we first compared antigen presentation by wild-type B cells and HEL-specific MD4 B cells. MD4 B cells efficiently presented HEL-derived peptides even at low antigen concentrations (Supplementary Fig. 5f). Wild-type and MD4 B cells were cultured with or without HEL antigen (1 μg/mL), mixed at a 1:1 ratio, and labeled with a violet tracking dye. The cell mixture was then adoptively transferred into wild-type recipient mice, which subsequently received antibody

treatment. Notably, HEL-stimulated MD4 B cells were significantly depleted in mice treated with iTab (Supplementary Fig. 5g). These results suggest that iTabs can induce antigen-dependent depletion of B cells in vivo.

To avoid HEL-specific T cell activation by FR⁺ HEL peptide immunization, we generated a FR⁺ HEL peptide in which HEL-specific TCR binding residues were mutated (HEL_{Y53A-L56A}) (Supplementary Fig. 4d)[20] and analyzed iTab production. Surprisingly, the FR⁺ HEL_{Y53A-L56A} peptide also induced iTabs similar to wild-type FR⁺ HEL peptide (Fig. 4a, b). Furthermore, the iTabs blocked 3A9 TCR-Fc binding (Fig. 4c). Next, we analyzed the effect of the FR⁺ HEL_{Y53A-L56A} peptide immunization on the response to wild-type HEL peptide. Mice were immunized with FR⁺ or FR⁻ HEL_{Y53A-L56A} peptide, followed by wild-type FR⁺ HEL peptide immunization. CD4⁺ T cells from FR⁺ HEL_{Y53A-L56A} peptide immunized mice showed reduced IL-2 production and proliferation compared with CD4⁺ T cells from FR⁻ HEL_{Y53A-L56A} peptide immunized mice (Fig. 4d). These data suggest that HEL-specific iTabs suppress the activation of HEL-specific T cell responses in vivo.

## The overall structure of the MHC-II−HEL peptide−iTab Fab ternary complex

We determined the cryo-EM structure of the MHC-II−HEL peptide−iTab (11−72) Fab ternary complex at a resolution of 3.09 Å (Fig. 5a). The data collection and refinement statistics are provided (Supplementary Fig. 6a−c and Supplementary Table 2). The structure of the MHC-II−HEL peptide−iTab Fab ternary complex reveals that the

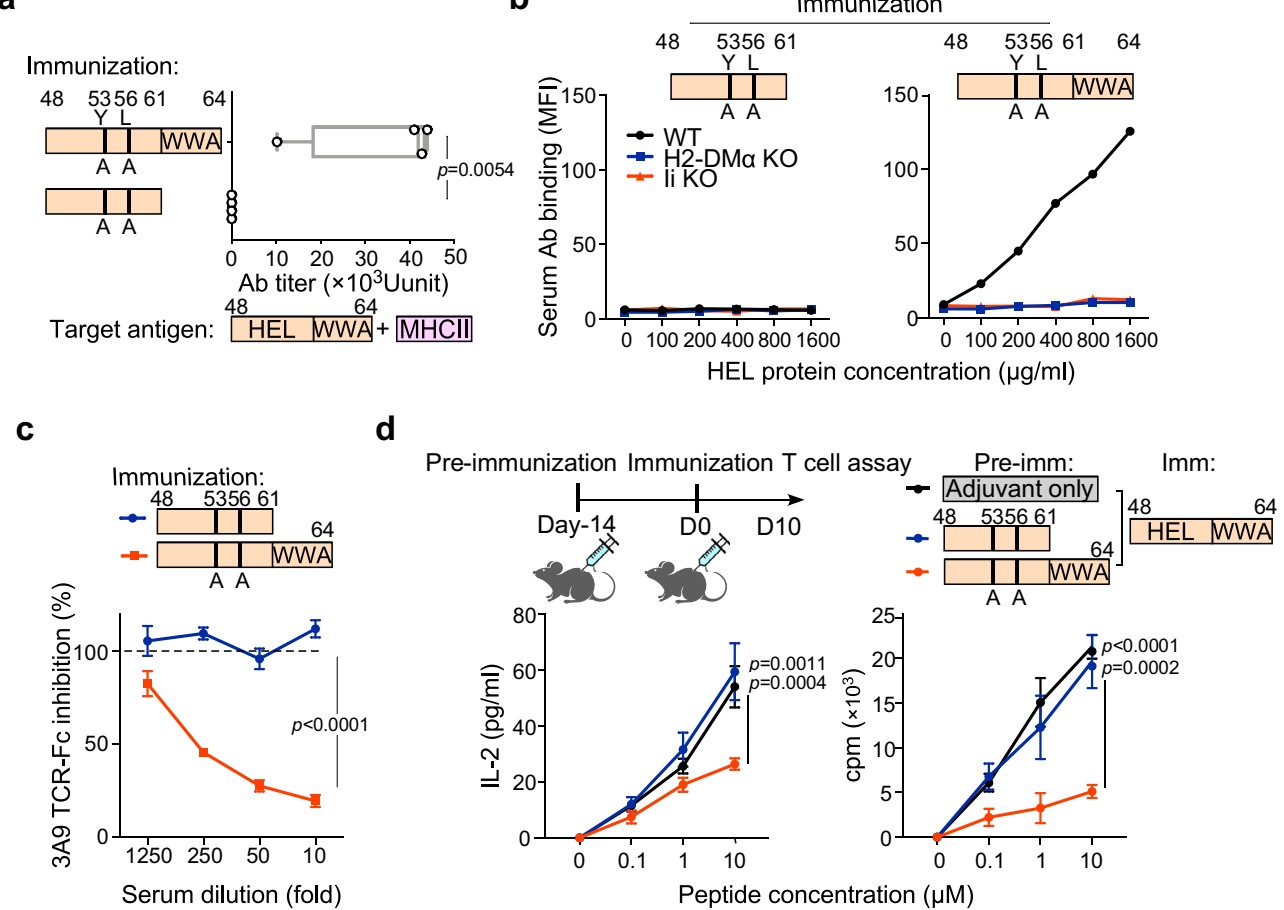

**Fig. 4 | Immunization of FR⁺ peptides with mutations in TCR recognition sites inhibit T cell response by inducing iTabs. a** Production of iTabs by immunization with mutated FR⁻ or FR⁺ HEL peptides ($n = 4$ mice per group). **b** Binding of serum Abs from mice immunized with mutated FR⁻ or FR⁺ HEL peptides against WT LK35.2(black line), H2-DMα (blue line) or invariant chain (red line) knockout cells pulsed with HEL protein. **c** Blocking of 3A9 TCR-Fc binding by Abs induced by mutated FR⁻ or FR⁺ HEL peptide immunization. **d** Inhibition of HEL-specific T cell response by immunization with mutated FR⁺ HEL peptide. Mice immunized with mutated FR⁻ or FR⁺ HEL peptide were further immunized with WT HEL peptides. IL-2 production (left) and T cell proliferation (right) upon FR⁺ HEL peptide stimulation were shown (**b,c** and **d**, $n = 3$ technical replicates). $p$-values were determined by two-way ANOVA for [c and d] with Tukey's correction, except for (**a**) where a two-sided Student's $t$ test was used. Bars represent means (min–max) (**a**) and mean values +/− SD (**c, d**). The data represent three independent experiments.

Fab fragment binds to the HEL peptide binding groove of MHC-II. The overall structure of MHC-II exhibits excellent superimposition with its previously reported structure (PDB code 1IAK[21],) (Fig. 5b). The root mean square deviation (r.m.s.d.) value for the main chain Cα atoms between the MHC-II–HEL peptide–iTab Fab ternary complex and the MHC-II–HEL peptide complex (PDB code 1IAK) is 1.076 Å.

**Binding interfaces for iTab and the MHC-II-HEL peptide complex**
Remarkably, the C-terminal FR of the HEL peptide is recognized by the Fab fragment, and its conformation within the MHC-II–HEL peptide–iTab Fab ternary complex differs from that observed in the previously reported structure of the MHC-II-HEL peptide binary complex (PDB code 1IAK[21],) (Fig. 5b). Our study elucidates the Fab binding modes of the C-terminal FR of the HEL peptide and MHC-II within the ternary complex.

First, to quantify interactions within the MHC-II–HEL peptide–iTab Fab ternary complex, we calculated buried surface areas (BSA) at each interface using AreaIMol (CCP4) based on the refined structural model (Supplementary Table 3). The BSA between the full HEL peptide and iTab Fab, between HEL residues W62–W63 and iTab Fab, and between MHC-II and iTab Fab were 592.3 Å², 301.4 Å², and 690.3 Å², respectively. Thus, W62 and W63 account for approximately

51% of the total peptide–Fab interface (301.4/592.3), highlighting the dominant contribution of the C-terminal WW motif to iTab binding.

Secondly, we investigated the 11–72 mAb binding modes of W62 and W63, the C-terminal FR of the HEL peptide, because peptide mutation analysis has demonstrated the essentiality of W62 and W63 for antibody binding (Fig. 6a). Within the complex, W62 of the HEL peptide is spatially proximate to Y107, S108, R114, Y115, and P116 of the light chain (L) and W52, N66, and Y113 of the heavy chain (H) (Fig. 5c). Similarly, W63 of the HEL peptide is in close proximity to Y107 of the light chain and Y38, S110, S111, Y112A, D112, and Y113 of the heavy chain. The indole ring of W63 forms a hydrogen bond with the main-chain carbonyl group of S110/H (Fig. 5d).

The MHC-II protein's α chain (I-Aᵏα) is recognized by the light chain of the Fab fragment (Fig. 5e). Specifically, T69 of I-Aᵏα (T69/α) forms hydrogen bonds with Q27/L, while H72/α engages in hydrophobic interactions with R114/L (Fig. 5e). Conversely, the β chain (I-Aᵏβ) interacts with both the light and heavy chains (Fig. 5f). The main chain of E59/β and the side chain of Y60/β participate in hydrophobic interactions with Y112/H. The side chain of K63/β forms hydrophobic interactions with W56/L and Y112/H. The side-chain amino group of Q64/β forms hydrogen bonds with the main-chain carbonyl group of Y112/H. Finally, the side chain of Y65/β establishes hydrophobic contacts with the main chain of V29/L and G36/L (Fig. 5f).

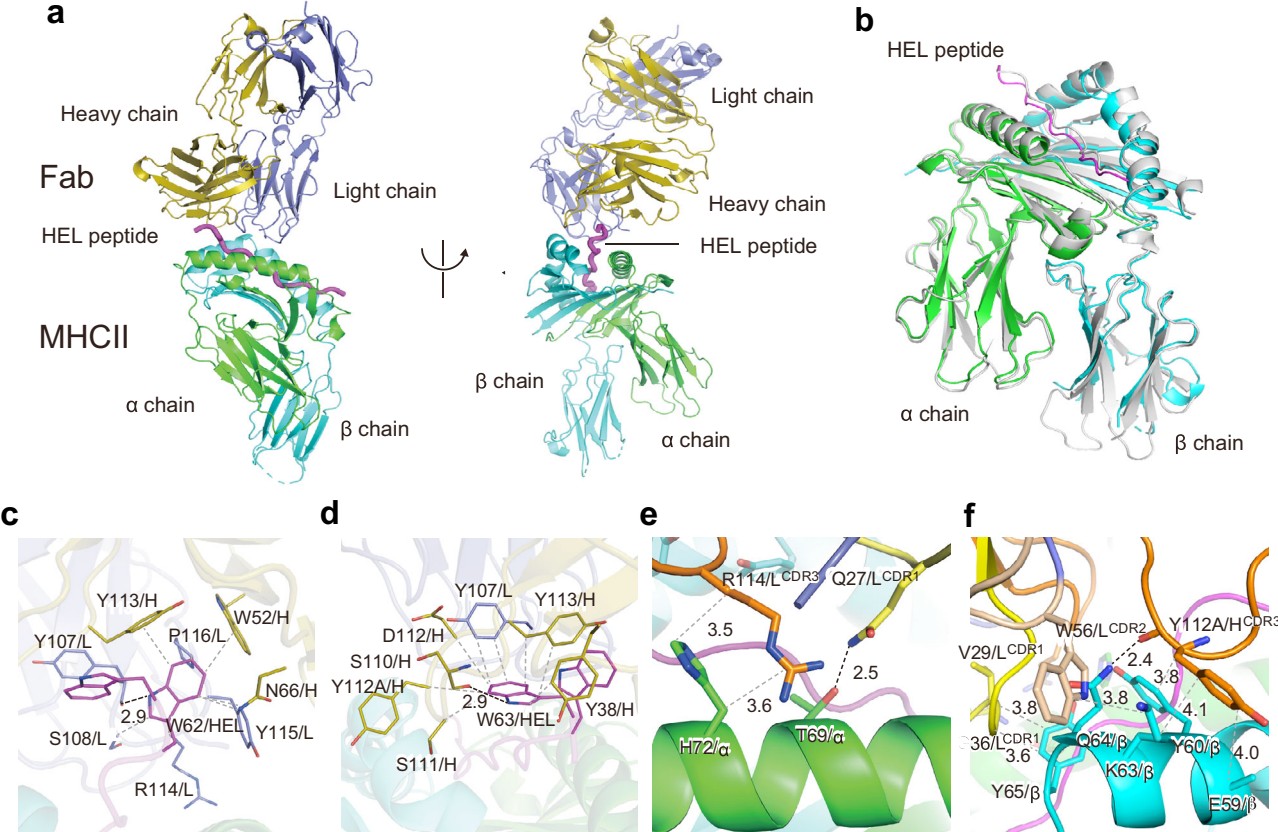

**Fig. 5 | iTab recognizes both peptide FR and MHC-II. a** Overall structure of the MHC-II-HEL peptide-iTab Fab ternary complex. **b** Comparison of the MHC-II structures. Superimposition of the MHC-II-HEL peptide-iTab Fab ternary complex and the murine MHC class II I-A$^k$ in complex with HEL peptide (PDB code 1IAK, gray). Fab fragment is omitted here for clarity. The α and β chains of MHC-II, heavy and light chains of Fab fragment, and HEL peptide are colored green, cyan, olive, slate blue, and magenta, respectively. **c, d** The interaction of W62 (**c**) and W63 (**d**) residues of the HEL peptide and Fab fragment. **e, f** The interaction of the α (**e**) and β (**f**)

chains of MHC-II and Fab fragment. Residues involved in the interaction are shown as stick models. The α and β chains of MHC-II, heavy and light chains of Fab fragment, and HEL peptide are colored green, cyan, slate blue, yellow, and magenta, respectively. CDR1, CDR2 and CDR3 of Fab fragments are colored yellow, wheat, and orange, respectively. The side chain of MHCβE59 is omitted because it was modeled as alanine. The amino acid residue numbers were assigned based on the IMGT information system (see Supplementary Table 4).

Based on these results, we conducted a mutation analysis of MHC-II and the 11−72 mAb at their interaction interface. MHC-II mutants demonstrated equal or enhanced presentation of FR$^+$ HEL peptides (Fig. 6b). Conversely, binding of the 11−72 mAb to Y60A/β and Y65A/β mutants was significantly reduced (Fig. 6c). The aromatic ring of tyrosine proved crucial for antibody recognition, as substitution with phenylalanine restored antibody binding. A similar pattern was observed with antibody mutants at Y38/H, Y113/H, and Y107/L (Fig. 6d). These results indicate that not all interaction sites necessarily impact antibody binding. Potential explanations include weak binding or compensatory interactions at other sites.

## iTabs suppress the development of autoimmunity

Experimental autoimmune encephalomyelitis (EAE) serves as an autoimmune disease model for multiple sclerosis, mediated by encephalitogenic CD4$^+$ T cells. EAE is induced in SJL/J mice by immunization with the autoantigen, proteolipid protein (PLP) peptide[22]. It has been reported that the FR$^-$ PLP peptide (PLP$_{139−151}$) induces chronic relapsing-remitting EAE more than the PLP with an N-terminal FR (PLP$_{136−150}$)[23].

We analyzed serum antibodies from mice immunized with an FR$^-$ PLP peptide (PLP$_{139−151}$) or an FR$^+$ PLP peptide (PLP$_{136−151}$) to determine whether iTabs were produced. Similar to the HEL or Sm-P40 peptide, anti-PLP iTab was induced in mice immunized with the FR$^+$ PLP peptide but not in mice immunized with the FR$^-$ PLP peptide (Fig. 7a).

Furthermore, immunization with a mutated FR$^+$ PLP peptide, in which the pathogenic TCR epitope (PLP$_{H147K}$) was mutated, also induced anti-PLP iTab (Fig. 7a)[24]. In contrast, anti-PLP peptide antibody titers were not significantly different between mice immunized with FR$^-$ PLP H147K and FR$^+$ PLP H147K (Supplementary Fig. 7a, b). A monoclonal anti-PLP iTab (clone 2036) obtained from FR$^+$ PLP peptide immunized mice did not show any binding to MHC-II only or to peptide only (Supplementary Fig. 7c, d). Activation of GFP-reporter cells expressing five pathogenic anti-PLP TCRs[22] was blocked by the monoclonal iTab except for SPL1.1 TCR (Fig. 7b). We also examined whether iTabs arise in a non-immunized autoimmune context. Serial serum samples collected from NOD mice during diabetes development did not show detectable iTabs recognizing the InsB9−25–I-A$^{g7}$ complex (Supplementary Fig. 7e).

To investigate the effect of anti-PLP iTabs on the development of EAE, we analyzed cytokine production in the culture supernatant of lymphocytes from FR$^+$ PLP peptide-immunized mice treated with anti-PLP iTabs. Lymphocytes co-cultured with the FR$^+$ PLP peptide from anti-PLP iTab-treated mice showed significantly reduced inflammatory cytokine production compared with lymphocytes from mice treated with the isotype control antibody-treated mice (Fig. 7c). When T cells from the anti-PLP iTab-treated mice were adoptively transferred to naïve SJL/J mice, T cells from the iTab-treated mice induced significantly milder EAE with delayed onset than T cells from the control mAb-treated mice (Fig. 7d). When EAE-induced mice were treated with

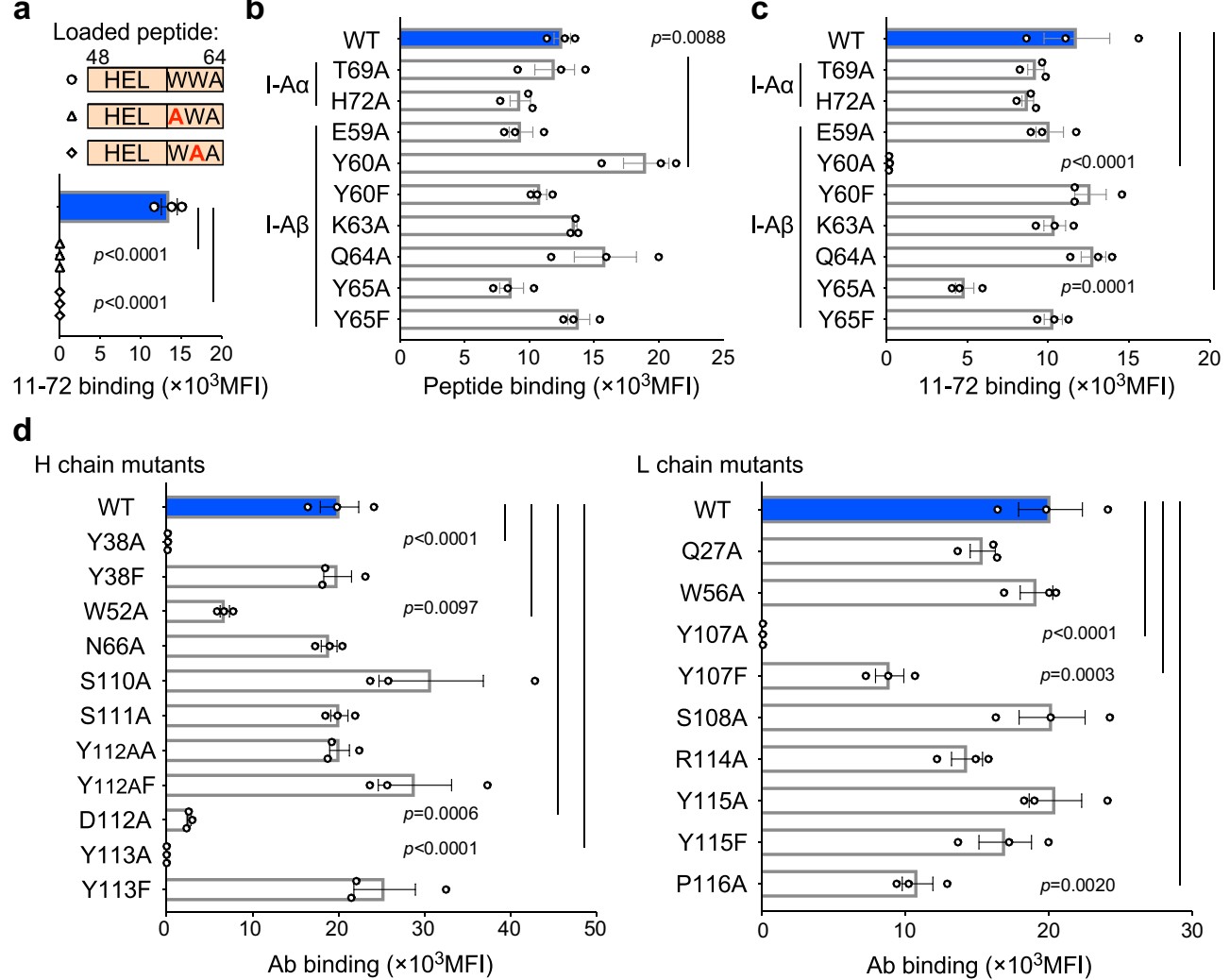

**Fig. 6 | Specific amino acid residues on peptide FR and MHC-II are recognized by iTab. a** Binding of 11–72 mAb to WT and mutated HEL peptides presented on MHC-II. **b** HEL$_{48-64}$ peptide presentation by MHC-II mutants. **c** 11–72 mAb binding to HEL$_{48-64}$ peptide presented on MHC-II mutants. **d** Binding of 11–72 mAb mutants to HEL$_{48-64}$ peptide-pulsed MHC-II. *p*-values were determined by one-way ANOVA with Dunnett's correction. Data represent the mean +/− SE (**a–d**) (*n* = 3 technical replicates). The experiments were replicated three times.

anti-PLP iTab, the iTab significantly suppressed the progression of EAE (Fig. 7e). These results suggest that self-antigen-specific iTabs are involved in the regulation of autoimmunity.

Finally, we examined whether immunization with iTab-inducing peptide could prevent EAE. Mice immunized with the FR$^+$ PLP$_{H147K}$ peptide, which induces iTabs without stimulating pathogenic T cells (Supplementary Fig. 7f), showed significantly milder EAE symptoms compared to FR$^-$ PLP$_{H147K}$ peptide immunized mice in which iTabs are not induced (Fig. 7f and Supplementary Fig. 7g). Despite similar anti-peptide antibody titers (Supplementary Fig. 7a), only immunization with the FR-containing mutated peptide resulted in suppression of EAE, suggesting that anti-peptide antibodies alone are unlikely to account for the observed suppressive effect. These results suggest that iTabs play a role in suppressing self-antigen-specific immune responses.

## Discussion

Although antibodies against the peptide-MHC-II complex can be produced by immunization with a purified peptide-MHC-II complex, it has not been known that iTabs are produced in a general immune response. In this study, we demonstrate that iTabs are produced during the immune response to several antigens in a T cell-dependent

manner. This observation suggests that iTabs represent an important yet previously underappreciated component of the adaptive immune response. Functionally, iTabs suppress CD4$^+$ T cell responses by blocking TCR recognition of peptide−MHC-II complexes and can also modulate antigen-presenting B cells via Fc-dependent effector functions, including ADCC; however, it remains unclear which mechanism is predominant in vivo. These activities suggest that iTabs may contribute to the contraction/resolution phase of antigen-specific immune responses and/or help restrain excessive immune activation, which can be detrimental under certain conditions (Supplementary Fig. 8).

The approach presented in this study is fundamentally distinct from classical hapten-carrier systems. In hapten-carrier immunization, an exogenous or synthetic hapten is chemically conjugated to a carrier protein, and the resulting antibody response is directed primarily against the hapten. In contrast, our work uses the endogenous MHC-II molecule as the physiological scaffold for antigen presentation. Moreover, iTabs recognize a composite, conformational epitope formed by the native peptide−MHC-II complex, closely mirroring the molecular features engaged by TCRs. This mode of recognition differs qualitatively from hapten-specific antibody responses and provides a mechanistically distinct framework for antigen-specific immune modulation.

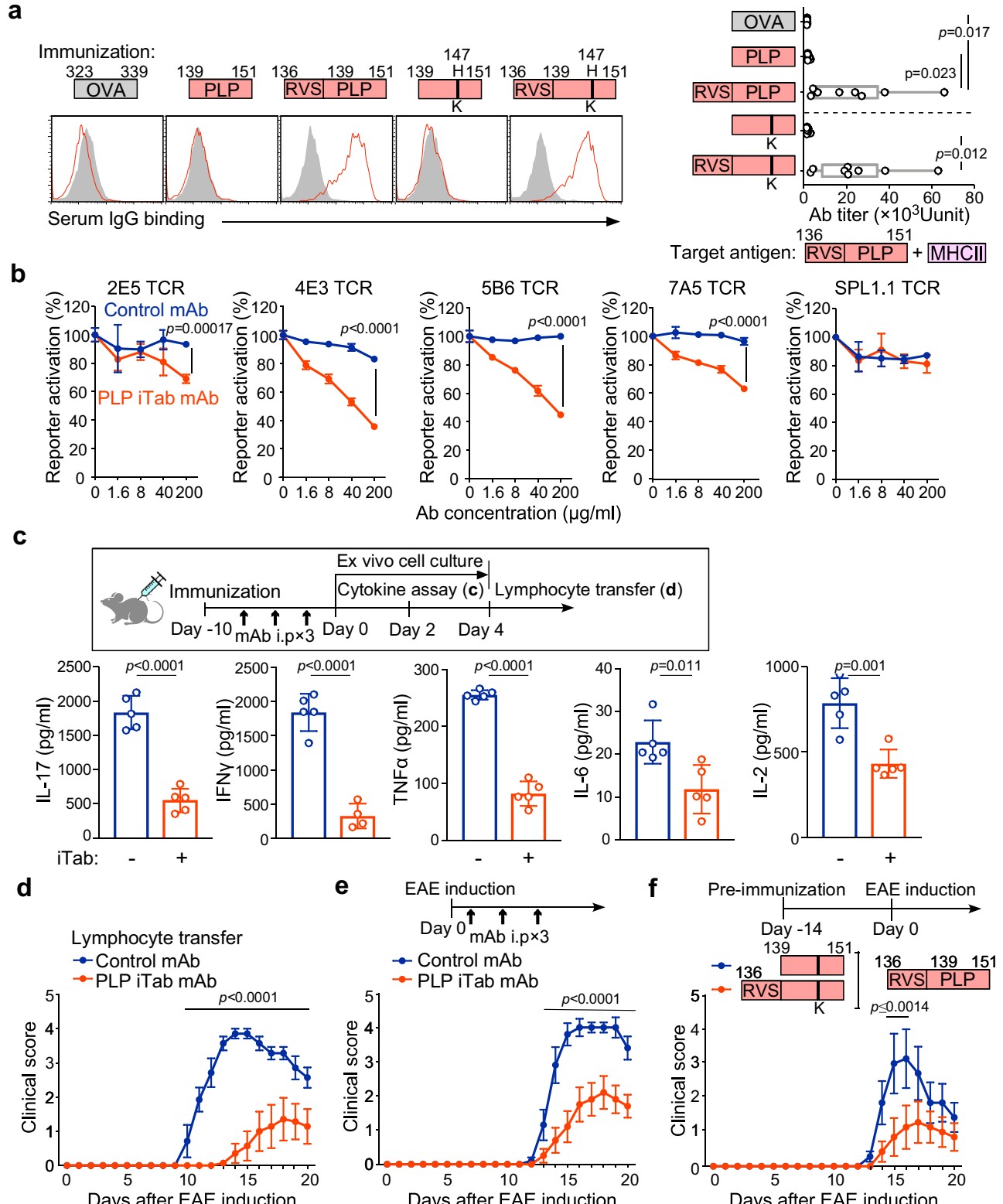

**Fig. 7 | iTabs induced by FR⁺ peptide immunization ameliorate EAE. a** iTab production by immunization with WT PLP, mutated PLP or OVA peptide. PLP-peptide pulsed cells were stained with immunized serum (red line). Control staining was shown as a shaded histogram (left). Serum Ab titers against PLP peptide-pulsed cells are shown (right) (*n* = 8 mice per group). **b** Monoclonal anti-PLP iTab, 2036, blocks the activation of NFAT-GFP reporter cells with PLP-specific TCRs in a dose-dependent manner (*n* = 3 technical replicates). **c** iTab-treated mice show decreased T cell response to FR⁺ PLP peptide immunization. Inflammatory cytokine production by T cells upon stimulation with PLP peptide was shown (*n* = 5 technical

replicates). **d** Clinical score of adoptive transfer EAE of ex vivo stimulated lymphocytes (*n* = 7 mice per group). **e** Anti-PLP iTab ameliorates PLP peptide-induced EAE (*n* = 10 mice per group). **f** Anti-PLP iTab-inducing peptide immunization ameliorates EAE (*n* = 7 mice per group). Data are presented as mean values (min−max) (**a**) and mean values +/− SD (**b**, **c**) and +/− SE (**d**−**f**). Statistical significance was tested by one-way ANOVA with Tukey's correction (**a**), and two-way ANOVA (**b**, **d**−**f**) with Bonferroni's correction, except for (**c**), where a two-sided Student's *t* test was used. All data are representative of three independent experiments.

FR at the N-terminus or C-terminus of antigen peptides are required for iTab production. In general, minimum peptides without FR have been used to analyze the CD4+ T cells response. Therefore, iTabs have not been observed in most peptide immunization studies. However, mass spectrometry analyses have demonstrated that many naturally processed peptides presented on MHC-II contain FR[14,15]. Although the specific amino acid residues involved in the production of iTab have not been identified, it is important to note that the immunogenicity of peptides varies greatly in the presence or absence of iTab-inducing FR.

Notably, iTab titers peaked at approximately two weeks after immunization and subsequently declined, in contrast to the more sustained accumulation often observed for conventional antigen-specific IgG responses. This transient kinetic profile suggests that iTab responses may be regulated differently from typical antibody responses to foreign antigens, despite evidence of T cell dependence and affinity maturation. One possible explanation is that iTabs decline because they are short-lived and/or are cleared relatively rapidly in vivo. We speculate that binding to cognate peptide-MHC-II complexes on antigen-presenting cells could contribute to their clearance, thereby linking iTab kinetics to the contraction/resolution phase of antigen-specific immune responses. The transient nature of their appearance may limit sustained suppressive activity but may also be consistent with a time-restricted regulatory role that attenuates excessive immune activation without broadly suppressing protective immunity.

From a therapeutic perspective, sustained iTab levels might be advantageous in chronic autoimmune settings; accordingly, boosting regimens and/or optimization of immunizing peptides could be explored to enhance the magnitude and durability of iTab induction. In therapeutic antibody discovery, improved monoclonal candidates are typically obtained through repeated antigen exposure and/or in vitro engineering followed by stringent screening to enrich for high-affinity variants. In the present study, hybridomas were generated from mice immunized only once, and we did not specifically select for the highest-affinity clones.

Structural analysis confirmed that iTabs recognize both FR and MHC-II. Analyses of iTab or MHC-II mutants show that iTabs recognize both FR and MHC-II. Importantly, peptides lacking FR did not induce iTabs, highlighting the critical role of C-terminal or N-terminal FRs in antigenicity for iTab induction. The binding sites of iTabs differed from those of previously reported TCR-like antibodies that recognize core peptide epitopes presented on MHC-II[5,8]. These previously reported TCR-like antibodies were generated by direct immunization with peptide-MHC-II complex, suggesting that these antibodies appear to be generated in a different manner than the iTabs.

Functionally, iTabs blocked most EAE-inducing pathogenic TCR recognition. Optimization of iTab-inducing antigens may lead to the development of more effective methods of specifically suppressing pathogenic autoreactive T cell activation. Finally, the lack of a robust therapeutic effect in the post-induction EAE setting may reflect intrinsic limitations of the model: clinical scores can begin to decline spontaneously before administered or induced antibodies reach sufficient levels to produce a measurable benefit, potentially obscuring treatment effects. In addition, no iTabs were detected in NOD mice, a spontaneous diabetes model. While this result does not allow definitive conclusions, it suggests that iTab production may not be a universal feature across autoimmune models and may depend on antigenic context or immune environment. Notably, insufficient induction of iTabs could plausibly contribute to dysregulated autoreactivity in some settings, a possibility that warrants further investigation.

Although this study did not address human applications due to the complexity of HLA polymorphism and the difficulty of obtaining clear human epitopes, our findings established a conceptual framework that offers insights into future strategies for antigen-specific immune modulation. Together, our data provide evidence that FR+ antigens can induce iTabs endogenously and support the concept that iTab induction may offer a framework for antigen-specific modulation of excessive immune responses in settings such as autoimmunity and allergy.

## Methods

### Mice

B10.A (B10. A/SgSnSlc H2$^a$), B10.D2 (B10. D2/nSnSlc H2$^d$), B10.S (B10. S/SgSlc H2$^s$), C57BL/10 (C57BL/10SnSlc H2$^b$) congenic, Balb/c (BALB/cCrSlc) and C57BL/6 J (C57BL/6JJmsSlc) mice were purchased from Japan SLC. SJL/J mice (Jackson strain no: 000686) were purchased from Charles River Laboratories Japan. NOD (NOD/SHiJcl) mice were purchased from CLEA JAPAN. To induce diabetes, cyclophosphamide (C2236, TCI) was administered to seven-week-old female NOD mice on day0 and day14[25]. MD4 BCR transgenic mice (Jackson strain no: 002595) were kindly provided by Prof. Tomonori Kurosaki (Osaka University). MD4 mice were backcrossed for over 15 generations to the B10.A-H2 h4 /(4 R) SgDvEgJ strain (Jackson strain no: 001150) purchased from the Jackson laboratory. Mice were kept in specific pathogen-free cages in a controlled environment, and experimental/control animals were co-housed. Mice were euthanized using carbon dioxide inhalation. All experiments were approved by the Animal Research Committee of the Research Institute for Microbial Diseases, Osaka University and conducted in accordance with the guidelines of the committee (R05-01-0).

### Cells

The human embryonic kidney 293 T cells (RCB2202) were purchased from the RIKEN Cell Bank, and LK35.2 cells (ATCC HB-98) expressing I-A$^k$ and NK92 cells (ATCC CRL-2407) were obtained from the American Type Culture Collection. The invariant chain (Ii) or H-2Mα deleted LK35.2 cells were generated by the CRISPR-Cas9 system using the pX330 vector (Addgene: ID42230) inserted into the primers (Ii:5′-caccgtacaccggtgtctctgtcc-3′ and 5′-aaacggacagagacaccggtgtac-3′, H-2Mα: 5′-caccgattcccaacatagggctct-3′ and 5′-aaacagagcccctatgttgggaatc-3′). The Expi293F cells were purchased from Thermo Fisher Scientific. Each cell line was tested regularly for *Mycoplasma* contamination using PCR.

### Plasmids and transfection

Plasmids for MHC class II molecules and 3A9 single-chain TCR-Fc fusion protein were constructed as previously described[12]. The cDNAs were cloned into the pME18S expression vector. MHC-II mutants were generated by site-directed mutagenesis. 293 T cells were transiently co-transfected with MHC-II and pMxs-GFP using PEI Max (24765, Polysciences), the peptide was added on day 2, and GFP-positive cells were analyzed 3 days post-transfection. The gating strategy of Ab titers and interaction analysis of monoclonal iTab and MHC-II mutants was shown in Supplementary Fig. 1b. The plasmids for the 3A9 single-chain TCR-Fc fusion protein were generated from cDNA derived from the 3A9 T cell hybridoma with 3 × (GGGS) linkers between the extracellular domains of the α and β chains. The construct was cloned into the pME18S expression vector, which contains the human IgG1 constant region. Plasmids for mouse CD4 (accession No.: NM_013488) and TCR α and β genes of 3A9, 2E5, 4E3, 5B6, 7A5 and SPL1.1 were synthesized (Integrated Device Technology) according to the published TCR gene sequence and cloned into pMxs retroviral vector[22,26]. The generation of these CD4 and TCRαβ stable transfectants was achieved with PLAT-E retroviral packaging cells with an amphotropic envelope. The sequence encoding 11–72 heavy and light chains from hybridoma cDNA was cloned into a pcDNA3.4 expression vector containing the SLAM signal sequence. Also, 11–72 heavy or light chain mutations were introduced using a QuikChange multi-mutagenesis kit (200514,

Agilent). Recombinant Human MOG (accession No.: HSU64567) fused with GS-linker and 6×His tag was also cloned into pcDNA3.4, then transfected to Expi293F cells. The DNA sequences of these constructs were confirmed by sequencing (ABI3130xl).

## Peptides

The HEL peptide library, overlapping by 10 residues, was purchased from SCRUM Inc. The 25-amino-acid OVA peptide library, overlapping by 8 residues, was purchased from GenScript. Other peptides were also obtained from GenScript, with a purity of > 90% as determined by high-performance liquid chromatography. The epitopes encoded in the different monomeric constructs used here include; $OVA_{323-339}$(ISQAVHAAHAEINEAGR), $HEL_{41-70}$(QATNRNTDGSTDYGILQ INSRWWCNDGRTP), $HEL_{41-61}$(QATNRNTDGSTDYGILQINSR), $HEL_{48-70}$ (DGSTDYGILQINSRWWCNDGRTP), $HEL_{48-61}$(DGSTDYGILQINSR), $HEL_{48-62}$(DGSTDYGILQINSRW), $HEL_{48-63}$(DGSTDYGILQINSRWW), $HEL_{48-64}$(DGSTDYGILQINSRWWA), $HEL_{48-64(62A)}$(DGSTDYGILQINSRA WA), $HEL_{48-64(63A)}$(DGSTDYGILQINSRWAA), $HEL_{48-64(53,54W)}$ (DGSTDWWILQINSRWAA), $HEL_{48-64(59,60K)}$(DGSTDYGILQIKKRWAA), $HEL_{48-64(59,60W)}$(DGSTDYGILQIWWRWAA), biotinylated $HEL_{48-64}$(BioG SGSDGSTDYGILQINSRWWA), Sm-$P40_{234-246}$(PKSDNQIKAVPAS), Sm-$P40_{237-246}$(DNQIKAVPAS), Sm-$P40_{237-249}$(DNQIKAVPASQAL), Sm-$P40_{234-246-WWA}$(PKSDNQIKAVPASWWA), $_{PKS}HEL_{52-61}$(PKSDYGILQ INSR), $_{PKS}HEL_{52-64}$(PKSDYGILQINSRWWA), $HEL_{48-61(53,56A)}$ (DGSTDA-GIAQINSR), $HEL_{48-64(53,56A)}$ (DGSTDAGIAQINSRWWA), $PLP_{136-151}$(RVS HSLGKWLGHPDKF), biotinylated $PLP_{136-151}$(BioGSGSRVSHSLGKWL GHPDKF), $PLP_{139-151}$(HSLGKWLGHPDKF), $PLP_{136-151(H147K)}$ (RVSHS LGKWLGKPDKF), $PLP_{139-151(H147K)}$ (HSLGKWLGKPDKF), $MOG_{35-55}$(MEV GWYRSPFSRVVHLYRNGK), $InsB_{9-25}$(SHLVEALYLVCGERGFF). We employed the $HEL_{48-64}$ peptide, in which the C-terminal cysteine was substituted with alanine, in order to inhibit the formation of S-S bonds in this paper.

## Immunization

Mice were immunized subcutaneously (s.c.) with 100 μg of HEL (L6876, Sigma-Aldrich), OVA (A5503, Sigma-Aldrich), recombinant MOG or 30 nmol of peptides in Complete Freund's adjuvant (F5881, Sigma-Aldrich). Pre-immunization with mutated peptides was conducted using Incomplete Freund's adjuvant (263910, BD Difco). Serum samples were collected from immunized six to eight-week-old female mice after a period of 2–4 weeks, and other immunization experiments also used the same mouse strain. B cell hybridomas were established according to standard protocols. Briefly, whole lymphocytes from lymph nodes were fused to P3U1 cells to generate hybridomas, which were then screened using MHC-II-transfectants pulsed with peptide by flow cytometry. The iTab clones were generated at a frequency of less than 1% of the total screened hybridomas. Antibodies from serum or mAbs were purified using Protein A and G sepharose (17127903 and 17061805, GE Healthcare) on a Profinia protein purification system (Bio-Rad) and gel filtration on an AKTA pure 25 (GE Healthcare)

## Flow cytometry

APCs were prepared by stimulating them with peptides or proteins for either an overnight period or 36 h. The cells or beads were stained with immunized mouse serum at a dilution of 1:100–1000 (cell) or 1:200–4000 (beads), and then reacted with allophycocyanin (APC)-conjugated anti-mouse IgG or Alexa Fluor 647-conjugated anti-mouse IgG1 for LK35.2 cell staining as a secondary antibody. Finally, the dead cells were stained with propidium iodide (PI). The Aw3.18 antibody was purified from the supernatant of a hybridoma culture (CRL-2826, ATCC). Antibody titers were determined by serial dilution of positive serum. Anti-HEL or HEL peptide antibodies in serum were removed by Streptavidin (SA) conjugated Sepharose beads (17511301, GE Healthcare) fused with biotinylated antigen. For the detection of anti-peptide antibodies or anti-HEL antibodies, SA (Z02043, Genscript) was labeled

with aldehyde/ sulfate latex beads (A37304, Invitrogen), and then biotinylated peptide or protein was mixed. The binding of the human receptor-Fc fusion 3A9 TCR-Fc was examined by premixed with an APC-conjugated anti-human IgG Fc antibody. The TCR-Fc competitive assay was conducted by reacting the diluted serum before the pre-mixed-TCR-Fc, followed by staining. APC-conjugated anti-mouse CD4 was employed in the reporter assay. For the depletion of CD4 T cells, anti-CD4 antibody (BE0003-1, BioXcell) (0.2 mg) was administered intraperitoneally (i.p.) on day -2. For intracellular staining, cells were fixed and permeabilized using the BD Cytofix/Cytoperm Fixation/ Permeabilization Solution Kit (554714, BD Biosciences), then stained with anti-mouse H2-DM, Ii (151002, Biolegend) and APC-conjugated anti-rat IgG. Flow cytometric analyses were conducted using either a FACSCalibur or a FACSVerse flow cytometer (BD Biosciences). The data were subsequently analyzed using FlowJo v10 software (FlowJo, LLC). For the details of antibodies used for flow cytometry, see Supplementary Table 5.

## Immunoprecipitation and mass spectrometry

The sample preparation for mass spectrometry was performed based on previous reports[27]. Briefly, HEL protein pulsed LK35.2 cells were lysed by a cell lysis buffer containing 20 mM Tris-HCl pH, 8.0, 1 % 3-[(3-cholamidopropyl) dimethylammonio]−1-propanesulfonate (CHAPS) (347-04723, DOJINDO), 5 mM ethylenediaminetetraacetic acid (EDTA), Protease inhibitor cocktail (P8340, Sigma-Aldrich), 0.1 mM iodoacetamide, and 1 mM phenylmethanesulfonylfluoride (PMSF) (27327-81, Nacalai, Japan) for 60 min at 4 °C on a rotator and cleared for 20 min of centrifugation at $21,000 \times g$. For immunoprecipitation, the cell lysates were incubated with biotinylated anti-mouse MHC-II (205303, Biolegend)-coupled SA Sepharose overnight at 4 °C on a rotator. The MHC-II antibody-coupled SA Sepharose capturing MHC-II-peptide complexes was washed six times with washing buffer 1 containing 250 mM NaCl, 50 mM Tris-HCl, pH 8.0 and six times with washing buffer 2 containing 50 mM Tris-HCl, pH 8.0. For the detection of peptides, the immunoprecipitated products were eluted using 10 % acetic acid (01-0280-5, Sigma-Aldrich). For analysis of MHC-II binding peptides, the eluted immunoprecipitated products were filtered through VIVASPIN 6, MWCO 10,000, PES (VS0601, Sartorius). The lyophilized peptides were reconstituted in 0.1% formic acid. After cleanup using a C18 tip column, the samples were subjected to LC-MS analysis. Mass spectrometric analysis was performed using a quadrupole time-of-flight mass spectrometer coupled with trapped ion mobility spectrometry (timsTOF Pro 2, Bruker Daltonics) equipped with a nanoHPLC system (nanoElute 2, Bruker Daltonics) and a nanocapillary C18 column (NTCC-360/75-3-105, Nikkyo Technos). Acquired raw data were analyzed using PEAKS Xpro (Bioinformatics Solutions Inc.), searching against HEL supplemented with a common contaminant database. Peptide identification was filtered using a score threshold of $−10\lg P \geq 15$.

## Reporter assay

The TCR reporter cells were mouse T cell hybridomas that had been stably transfected with NFAT-GFP and FLAG-tagged DAP12, as previously described[28,29]. The original hybridoma TCR αβ chains were depleted using the CRISPR-Cas9 system with the pX330 vector inserted into the primers (α chain; 5′-CACCGTGCCGAAAACCATG-GAATC-3′ and 5′-AAACGATTCCATGGTTTTCGGCAC-3′, βchain; 5′-CACCGAGAAATGTGACTCCACCCA-3′ and 5′-AAACTGGGTGGAGT CACATTTCTC-3′). Following cell depletion, mouse CD4 was stably transfected into the cells, and each TCR was introduced into the CD4-positive cells. The cells were sorted using a Cell Sorter (SH800Z, Sony). The TCR reporter cells (96-well plate at $2 \times 10^4$ cells per well) were co-cultured with HEL protein-pulsed LK35.2 cells or $PLP_{136-151}$ peptide-loaded 293 T cells transfected with I-A$^s$ ($4 \times 10^4$ cells per well) and supplemented with or without antibodies for 16 h. The

expression of GFP was analyzed using flow cytometry (gating strategy of reporter assay was shown in Supplementary Fig. 4d).

## In vitro proliferation and IL-2 secretion assays

CD4$^+$ T cells ($2 \times 10^5$ cells per well) were isolated from peripheral lymph nodes of peptide-immunized mice using BioMag anti-mouse IgG (310007, QIAGEN) and a mouse CD4 T cell isolation kit (130-104-454, Miltenyi Biotec). Splenocytes as APC ($6 \times 10^5$ cells per well) from wild-type mice were treated with Ack buffer and subsequently treated with mitomycin C. The cells were incubated with HEL$_{48-64}$ peptides for 72 h in Advanced RPMI 1640 (49140-15, Nacalai tesque) supplemented with 1% FBS (Hyclone), Glutamax (35050061, Invitrogen) and penicillin/streptomycin (26253-84, Nacalai tesque), and pulsed with 1 µCi ($^3$H) thymidine for 18 h before harvesting and reading on a beta counter reader (NET027, Perkin Elmer). Supernatants were collected 24 h later for the measurement of IL-2 by ELISA ready-set-go reagent set (88−7024, eBioscience).

## Delayed-type hypersensitivity assays

Ten days post-immunization of HEL$_{48-64}$ peptide, mice were challenged with 30 nmol of the same peptide, which was injected intradermally into the footpad. The right footpad was injected with the peptide, while the left footpad was injected with vehicle (PBS) as a baseline control. The degree of swelling was quantified 24 h post-challenge using a Thickness Gauge (SM-112, TECLOCK). Monoclonal antibodies (1 mg per injection) were administered intraperitoneally (i.p.) on days 3 and 6.

## Antibody-dependent cellular cytotoxicity assays

NK92[30,31] cells stably transfected with chimeric FcR (mouse FcγRIV extracellular domain and human FcγR III intracellular domain) (96-well V-bottom plate at $2 \times 10^5$ cells per well) were co-cultured with or without HEL$_{48-64}$ peptide-loaded 293 T cells transfected with I-A$^k$ ($2 \times 10^4$ cells per well) and supplemented with antibodies for 6 h. The ratio of PI-positive dead cells was analyzed using flow cytometry. In vivo assay, B cells were isolated from the spleen of eight to ten-week-old female B10.A(4 R) and B10.A(4 R)-MD4 mice using the mouse B cell isolation kit (130-090-862, Miltenyi Biotec). These B cells were co-cultured with 10ng/ml IL-4 (404-ML-010, R&D Systems), 5 µg/ml anti-CD40 antibody (102812, Biolegend), 5 µg/ml anti-IgM antibody (115-006-075, Jackson Immuno Research) with or without HEL protein for 18h. These cells were mixed at a 1:1 ratio, labeled using the Cell trace violet proliferation kit (C34571, Invitrogen), and then transferred into eight-week-old female wild-type B10.A(4 R) mice. After 30 min, 0.5 mg of antibody was administered intravenously. Three days later, labeled$^+$ cells were analyzed by flow cytometry. Anti-mouse IgMa antibody was used to distinguish wild-type B cells (labeled$^+$, IgMa$^-$) and MD4 B cells (labeled$^+$, IgMa$^+$). The gating strategy was shown in Supplementary Fig. 5h. The ratio of MD4 B cells to wild-type B cells was calculated. This ratio was further adjusted by setting the average ratio in mice without co-culture with HEL, treated with control antibody as 1.0.

## Sample preparation for cryoEM analysis

The construction of soluble MHC class II molecules has been previously described[32]. The I-A$^k$α plasmids were fused with a GSSGSSG-linker, TEV protease cleavage site, leucine zipper and His-tag at the C-terminus, and HEL$_{48-64}$ peptide was bound to the I-A$^k$β plasmids with a GGGGSLVPRGSGGGGSGS-linker, which was attached via a GSSGSSG-linker, TEV protease cleavage site, leucine zipper and a Flag-tag at the C-terminus. The plasmids were subsequently inserted into the pcDNA3.4 expression vector. The recombinant protein was obtained via the Expi293 expression system. Monoclonal antibodies were obtained from hybridoma cells by digestion to the Fab fragment using immobilized papain (20341, Sigma-Aldrich). The purified HEL peptide-MHC-II protein by Ni Sepharose excel (17371201, cytiva) was mixed

with the Fab fragment, and incubated on ice for 30 min. The mixture was loaded onto a Superdex 200 10/300 column (cytiva) equilibrated with 20 mM Tris-HCl buffer (pH 8.0) containing 150 mM NaCl and eluted with the same buffer. Fractions containing the complex were collected and concentrated to approximately 4 mg/ml by ultrafiltration.

## Cryo-EM data collection

Holey carbon film-coated copper grids (Quantifoil R1.2/1.3 Cu 200 mesh; Microtools GmbH) pretreated by Au sputtering were glow-discharged for 10 s using an ion coater (JEC-3000FC; JEOL). A solution of 4.0 µL (1.0 mg/mL) of the MHC-II–HEL peptide–Fab complex, suspended in a buffer containing 20 mM Tris-HCl (pH 8.0) and 100 mM NaCl, was applied to the grid and blotted with filter paper for 6 s. The grid was then immediately plunged into a bath of cooled ethane using a FEI Vitrobot Mark IV (Thermo Fisher Scientific) under 100% humidity at 4 °C. Subsequently, the grids were examined using a CRYO ARM 300 electron microscope (JEOL)[33], equipped with a cold-field emission gun and an in-column energy filter with a slit width of 20 eV. Dose-fractionated images were recorded using a K3 Summit direct electron detector (AMETEK) in counting mode, with a dose rate of ~ 99 e$^-$ Å$^{-2}$ across 100 frames. All images were collected using SerialEM[34] with AI-assisted hole detection via yoneoLocr[35], at a nominal magnification of 100,000 ×, corresponding to a pixel size of 0.495 Å. The defocus range was estimated to − 0.8 µm to − 5.0 µm. A total of 17,175 movies were acquired.

## Cryo-EM data processing

Data processing was performed using CryoSPARC 4.2.0[36] and Relion 3.1.4[37]. The collected movies were aligned, dose-weighted, and averaged through motion correction implemented in CryoSPARC. The contrast transfer function (CTF) was estimated using Ctffind-4.1.13[38]. Subsequently, micrographs with an estimated resolution below 10 Å were excluded from the data processing. A total of 1,327,931 particles in 12,394 micrographs were selected through Blob picker and subjected to the first two-dimensional (2D) classification, template picker, and second 2D classification. Then, an initial three-dimensional (3D) map (C1 symmetry) was reconstructed through ab-initio reconstruction in CryoSPARC. Following homogeneous and non-uniform refinement yielded a 3D map at 3.7 Å resolution. The micrographs and 3D map obtained were exported to RELION 3.1.4, where particles were picked up again using Topaz and 3D reference picking due to a tendency for preferred orientations of the particles. Subsequent 2D and 3D classification in RELION yielded a total of 786,526 particles. Several rounds of 3D refinement, CTF refinement, and Bayesian polishing resulted in a 3D map at 2.91 Å resolution. However, the obtained 3D map was highly disordered, making model building difficult. Therefore, the selected particles and the 3D map were imported back into CryoSPARC and subjected to 3D classification (20 classes) again, resulting in the selection of 50,032 high-quality particles. Additionally, 3D flexible refinement (3DFlex)[37] improved the disorder of the 3D map, resulting in a 3D map at 3.09 Å resolution based on the gold-standard Fourier Shell Correlation cutoff (0.143) criterion, through post-processing in RELION. Furthermore, an enhanced map was generated using EMready[39], a program for improving the interpretability of the 3D map by leveraging similarity and correlation-guided deep learning, which was used as a reference for model building.

## Atomic model building

An initial model was generated using the structure of murine MHC-II class II I-A$^k$ with a peptide from hen egg lysozyme (PDB code: 1IAK[21]) and H-2 class II histocompatibility antigen (PDB code: 2PXY[40]) through the SWISS-MODEL server (https://swissmodel.expasy.org/). These models were fitted into the map using UCSF Chimera[41]. The fitted initial model was manually adjusted with COOT[42] to better fit the 3D map,

followed by refinement using Phenix[43] and REFMAC5 in CCP-EM[44,45]. The refinement statistics of the final model were obtained using the comprehensive validation program in Phenix.

### Cytokine secretion assay and adoptive transfer of EAE

Female SJL/J mice, aged 10–14 weeks, were immunized s.c. with 100 µl of a CFA emulsion containing 200 µg of *Mycobacterium tuberculosis* H37Ra (231141, Difco Laboratories) and 15 nmol of $PLP_{136-151}$ peptide. Monoclonal anti-PLP iTab (1 mg per injection) was administrated i.p. on days 0,4 and 8. Cytokine release was evaluated by culturing lymphocytes from lymph nodes on day 10 in vitro with 0.1 µM of $PLP_{136-151}$ peptide or an irrelevant peptide ($OVA_{323-339}$) for 48 h. Supernatants were collected and analyzed using a mouse Th1/Th2/Th17 cytokine kit (560486, BD Bioscience) by flow cytometry. Lymphocytes were cultured for an additional 48 h, after which $1 \times 10^7$ cells were transferred to naïve SJL/J mice. The animals were evaluated daily for clinical signs of disease as follows: (0) No disease; (0.5) Loss of tail tonus; (1) Complete tail paralysis; (2) Tail paralysis with hind limb weakness; (3) Complete hind limb paralysis; (4) Hind limbs paralyzed, weakness in forelimbs; (5) Tetraplegia; (6) Death.

### iTab or peptide therapy in EAE

EAE was induced by immunization with 100 µl of a CFA emulsion containing 250 µg of *Mycobacterium tuberculosis* H37Ra and 50 nmol of $PLP_{136-151}$ peptide. In iTab therapy, Monoclonal anti-PLP iTab (1 mg per injection) was administered i.p. on days 0,4 and 8. Two weeks before the onset of EAE, a pre-immunization procedure was conducted in which 50 nmol of PLP peptide, which mutated for the pathogenic TCR recognition site at H147K, was administered s.c. with IFA.

### Statistical analysis

Data were compared by Student's *t*- test, one-way or two-way ANOVA tests. *p*-values < 0.05 were considered statistically significant.

### Reporting summary

Further information on research design is available in the Nature Portfolio Reporting Summary linked to this article.

## Data availability

Atomic coordinates and cryo-EM maps for the reported structure of anti-HEL iTab were deposited in the Protein Data Bank under accession code 9L1L, and in the Electron Microscopy Data Bank under accession code EMD-62748. The mass spectrometry data have been deposited in the Japan ProteOme Standard Repository (jPOSTrepo)[46] under the accession code JPST004463. All data are included in the Supplementary Information or available from the authors, as are unique reagents used in this Article. The raw numbers for charts and graphs are available in the Source Data file whenever possible. Source data are provided in this paper.

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

## Acknowledgements

We are grateful to Rena Yanagisawa, Kaori Oshimo, Asa Tada and Yumi Inaba for their technical assistance. We thank Hiroko Kato and Ryohei Narumi for the mass spectrometry. This work was supported by JSPS KAKENHI under Grant Number 22H04989 (H.A.), and the Japan Agency for Medical Research and Development (AMED) under Grant Numbers JP25ek0410124 (H.A.), JP25gm1810006 (H.A.), and JP223fa627002 (H.A.), JP25ama121006 (Ke.K., K.Y.), and Japan Science and Technology Agency (JST) Mirai Program under Grant Number JPMJMI23G2 (K.Y.).

## Author contributions

K.K. performed most of the experiments, analyzed and discussed the data. W.N. discussed the data and assisted with experiments. Ke.K. and K.Y. screened frozen-hydrated samples on the EM grids and collected cryo-EM data. Ke.K. performed EM data processing, structure analysis, model building and structure interpretation. H.T. and S.Y. performed protein sample preparation for cryo-EM, and structural data analysis and discussion. H.A. designed the study. All authors contributed to the writing of the manuscript.

## Competing interests

The authors declare no competing interests.
