## [Transparent Peer Review file · Nature Communications]

Immune-induced TCR-like antibodies regulate specific T cell response in mice

Corresponding Author: Professor Hisashi Arase

Version 0:

Reviewer comments:

Reviewer #1

(Remarks to the Author)

In this manuscript, Kishida et al report on a previously undescribed class of "TCR-like" antibodies that recognize a specific MHC/peptide complex. These antibodies form in immunized mice and appear to require the presence of a large flanking amino acid residue (outside the minimal MHC-binding nonamer). The authors demonstrate that such "TCR-like" antibodies are formed in a few different settings and that the key flanking amino acid can be an N-terminal or C-terminal extension of the minimal peptide. For one particular PLP peptide, transfer of "TCR-like" antibody reduced the severity of EAE. In total, these observations are novel and interesting, but some aspects of the work are conceptually troubling:

- 1) The authors clearly state that formation of these "TCR-like" antibodies is transient and that they disappear from the antibody repertoire within a matter of weeks. As such, it is likely that these antibodies are not affinity matured and should not be considered to be part of the natural immune response.
- 2) This concern is compounded by the fact that most of the experiments rely on peptide immunizations. It could be argued that the MHC complex acts in a similar manner to a hapten. It is well known that haptens can be used to generate antibodies against small molecules including drugs, toxins, or even chemical groups - including modified amino acids. This undermines the novelty and relevance of the presented work. Limited experiments did utilize a single protein for immunizations and these appeared to generate lower titers of "TCR-like" antibodies, but it remains unclear whether a typical vaccine would generate "TCR-like" antibodies.
- 3) Although the authors present data showing that "TCR-like" antibodies block the binding of a previously described antibody that recognized the specific MHC/peptide complex that they are studying, it could be argued that they never directly show that the antibody prevents T cell activation.
- 4) It is not clear whether all appropriate experimental controls have been performed. In particular, did the authors search for "TCR-like" antibodies in non-immunized animals. These could be present if they are germline antibodies (this would also explain their disappearance as the antibody response matures)
- 5) When an antigen is naturally processed and presented via the MHC class II presentation pathway, it is clear from peptidomics data sets that the resulting "epitopes" exist as nested sets of peptides with varying numbers of n-terminal and c-terminal flanking residues. Thus, it would be helpful if the authors could present some evidence that their peptide sequences of interest are naturally processed and presented. In addition, there is some complexity with respect to which HLA/peptide complexes will be present on the surface of antigen presenting cells that have taken up and presented the antigen of interest and which of these would be bound by "TCR-like" antibodies.
- 6) Beyond this, assuming that "TCR-like" antibodies do bind to a given antigen presenting cell, would the most potent biologic effect be to block T cell activation or would this trigger antibody-dependent cellular cytotoxicity and death of the antigen presenting cell? The authors should probably do at least some work to verify this mechanism.
- 7) It could be argued that the degree of clinical improvement in the EAE model is not that impressive. Do these "TCR-like" antibodies really provide a promising therapy for autoimmune disease?

8) The discussion should include more critical thinking about the experimental results and potential limitations of the experimental system used. Most notably, given the absence of physiologic immunization and the decline in the levels of "TCR-like" antibodies by two weeks after immunization, it is not clear how much a part of the "natural" immune response these are. Thinking about these antibodies as a potential therapeutic, it seems likely that additional refinement of the antibody sequence to improve its affinity would be needed to create a monoclonal that could be put forward as a drug candidate. Could this be done without compromising their specificity?

Reviewer #2

(Remarks to the Author)

This study reports the novel discovery that immunization with protein antigens, such as HEL or peptides containing flanking residues (FRs) can induce the production of TCR-like antibodies (iTabs) in mice. These iTabs specifically recognize peptide-MHC class II complexes and block TCR engagement, offering a potential strategy for suppressing antigen-specific CD4⁺ T cell responses. The authors determined a cryo-EM structure of the iTab Fab in complex with the HEL-MHC II complex, revealing the detailed iTab recognition of the peptide, the C-terminal FR of the peptide, and the MHC II molecule. Overall, the findings are innovative, well-supported by experimental evidence, and provide a promising new approach for controlling autoimmune diseases.

Specific comments:

1. Lines 89-90, "To analyze the role of C-terminal and N-terminal FR in inducing iTabs, we generated recombinant HEL or Sm-P40 peptides containing C-terminal WWA FR or N-terminal PKS FR." The authors may want to explain the rationale for choosing the WWA and PKS residues to elicit the antibody response.
2. Can the authors report the buried surface area values of the antibody interface with the peptide (excluding the C-terminal WW), the C-terminal WW residues, and the MHC-II molecule?
3. Line 371 "The construction of soluble MHC class II molecules has been previously described." Please provide the appropriate references here.
4. Fig 5b legend, the PDB code 1IAK should be changed to 1IAK.
5. It would be helpful if the authors could change the antibody numbering to a standardized numbering scheme, such as IMGT, Kabat, or Chothia.
6. Fig 5f: The current color scheme uses pink and magenta, which are difficult to distinguish. The authors may want to change the colors to improve visual clarity. In addition, please include the chain IDs for K63 and Y60. In the figure, E59 appears to be an alanine.

Reviewer #3

(Remarks to the Author)

Review of Kishida et al
April 30, 2025

The central question of the study was whether peptide:MHCII antibodies are generated during the immune response, and if these antibodies serve as negative immune regulators to limit T cell effector function. The authors also asked whether longer peptides were more effective at generating p:MHCII antibodies compared to short peptides lacking MHCII flanking residues.

The central question has broad appeal to the immunology and medical community yet the rationale behind this question is not well supported by observations in humans or mice. While the authors provide compelling data that p:MHCII antibodies are generated following immunization with HEL and PLP-derived peptides, the data does not robustly support the authors claims that these antibodies serve as a negative regulator of T cell immunity. Furthermore, the authors claim that pMHCII antibodies generated during peptide immunization could potentially be used to control autoimmune diseases but the authors do not specifically discuss how this may be applied to human health. Many studies have used pMHCII antibodies as biologics in preclinical models of human disease so that is not novel. How do the authors envision peptide immunization to prevent or treat autoimmune disease?

Major points:

- 1) Immunizing mice with mutant peptide, that cannot activate T cells, prior to wild type peptide immunization, leads to a reduction in T cell proliferation and cytokine production (Fig 4d) and EAE disease (Fig 7f). The authors claim that reduced T cell proliferation and effector function, and EAE disease, is a direct result of endogenous production of pMHCII antibodies. As an alternative explanation, could the initial peptide immunization generate anti-peptide (not pMHCII) specific antibodies that then neutralize the wild type peptide, preventing it from being taken up by APCs and being efficiently presented to T cells? Can the authors show by ELISA whether HEL or PLP peptide specific antibodies are generated by immunization? For the experiment described in figure 7c, can serum from immunized mice be used to suppress T cell responses instead of purified antibody? If so, perhaps the serum could be used to determine if reduced T cell responses result from pMHCII antibodies vs peptide antibodies. I wonder if the immune serum could be absorbed using peptide prior to transfer to eliminate the peptide specific antibodies, leaving the pMHCII antibodies in the serum?
- 2) Supplemental figure 7 shows a model whereby pMHCII antibodies are involved in the contraction phase of the immune response. Can the authors show antigen specific T cell expansion kinetics in WT vs BCR transgenic mice to support the idea that antibodies are helping to resolve the inflammatory process, creating a steeper or earlier drop in T cell number

following expansion.

3) The authors use two model antigens, HEL and PLP, to draw broad conclusions about the need for flanking residues in pMHCII antibody generation. Structural analysis of pMHC antibodies has shown that these antibodies can have several modes of binding antigen, including TCR-like footprint across the pMHC (PMID: 36593402). To make the claim that pMHC antibodies are generally directed against flanking residues I suggest the authors include a wider range of peptides, especially peptides with bulky or charged residues in the MHC groove but exposed for TCR binding, like 2W or 3K.

4) Little information was provided about the frequency of B cells that are pMHCII-specific and whether they underwent T dependent affinity maturation prior to generating hybridomas. What fraction of hybridomas screened were pMHCII specific? The significance of the work would be enhanced by a more thorough description of the frequency of pMHCII-specific B cells before and after immunization, how this changes depending on the peptide epitope, and strain of mice. The translational impact would also benefit by determining whether pMHCII antibodies naturally exist in other autoimmune disease models, such as anti-InsB:IAg7 in NOD mice (PMID: 24550292) or anti-citFib:DR4 in DR4-IE mice (PMID: 18391064). Furthermore, the authors could use pMHC tetramers to screen human PBMC for pMHC antibodies against viral antigens induced by vaccination.

Minor points:

1) A major conclusion is based on the experiment shown in Fig7f yet the results are not very robust, with a modest, yet significant, reduction in peak disease. It states that this experiment was representative of 3. I suggest the authors show all 3 experiments in the supplement so help support their conclusion that pMHCII antibodies can impact autoimmune disease.

2) The introduction included results and a conclusion statement rather than a thorough explanation of the rationale for conducting the research. Overall, the writing would benefit from further editing for clarity and formatting.

Version 1:

Reviewer comments:

Reviewer #1

(Remarks to the Author)

The additional experimental data and revisions are sufficient to allay my concerns.

Reviewer #2

(Remarks to the Author)

The authors have addressed all my concerns.

Reviewer #3

(Remarks to the Author)

The authors have sufficiently addressed all of my concerns. I recommend the manuscript for publication.

Point by point response to the Reviewer's comments.

Reviewer #1:

General comment

In this manuscript, Kishida et al report on a previously undescribed class of "TCR-like" antibodies that recognize a specific MHC/peptide complex. These antibodies form in immunized mice and appear to require the presence of a large flanking amino acid residue (outside the minimal MHC-binding nonamer). The authors demonstrate that such "TCR-like" antibodies are formed in a few different settings and that the key flanking amino acid can be an N-terminal or C-terminal extension of the minimal peptide. For one particular PLP peptide, transfer of "TCR-like" antibody reduced the severity of EAE. In total, these observations are novel and interesting, but some aspects of the work are conceptually troubling:

Response to general comment:

We thank the reviewer for their careful evaluation of our manuscript and for recognizing the novelty and significance of our findings. We have carefully considered the conceptual concerns raised and performed additional experiments to address them. Our detailed responses to each comment are provided below.

Reviewer's comment 1

The authors clearly state that formation of these "TCR-like" antibodies is transient and that they disappear from the antibody repertoire within a matter of weeks. As such, it is likely that these antibodies are not affinity matured and should not be considered to be part of the natural immune response.

Response to this comment:

We agree with the reviewer that the kinetics of the TCR-like antibody response differ from those of long-lived antibody responses. However, transient production does not necessarily preclude affinity maturation or participation in an antigen-driven adaptive immune response *in vivo*. In our study, TCR-like antibodies were of the IgG isotype and were induced following antigen immunization, consistent with a conventional adaptive antibody response.

To directly assess T cell dependence, we depleted CD4⁺ T cells using anti-CD4 antibodies prior to immunization with HEL protein. Under these conditions, TCR-like antibody production was completely abolished (**new Supplementary Figure 2c**), demonstrating

that their generation is strictly CD4⁺ T cell-dependent. Moreover, sequence analysis of 11-72 monoclonal Ab revealed somatic mutations in both heavy (G36A, S57R) and light chains (A32T, S114R), consistent with T cell-dependent affinity maturation.

Taken together, these findings indicate that TCR-like antibodies are generated as part of an antigen-driven, T cell-dependent adaptive immune response, despite their transient appearance in the antibody repertoire.

These new data and interpretations are now described in the Results (page 5, lines 81–88) and Discussion (page 15-16, lines 332–342).

Reviewer's comment 2.

This concern is compounded by the fact that most of the experiments rely on peptide immunizations. It could be argued that the MHC complex acts in a similar manner to a hapten. It is well known that haptens can be used to generate antibodies against small molecules including drugs, toxins, or even chemical groups - including modified amino acids. This undermines the novelty and relevance of the presented work. Limited experiments did utilize a single protein for immunizations and these appeared to generate lower titers of "TCR-like" antibodies, but it remains unclear whether a typical vaccine would generate "TCR-like" antibodies.

Response to this comment:

We appreciate the reviewer's thoughtful comparison to classical hapten-carrier systems. While hapten-based immunization can indeed elicit antibodies against small molecules or chemical moieties, we believe that the mechanism and immunological context described in our study are fundamentally distinct.

In classical hapten systems, an exogenous or synthetic hapten is chemically conjugated to a carrier protein, and the resulting antibody response is directed primarily against the hapten itself. In contrast, our work focuses on the physiologically relevant MHC class II molecule as an endogenous platform for antigen presentation. The TCR-like antibodies described here recognize highly specific conformational epitopes formed by the native peptide-MHC-II complex, closely mirroring the molecular footprint of TCR recognition. Thus, unlike hapten-specific antibodies, these antibodies are directed against a composite, conformational structure that does not exist outside the context of antigen presentation.

To address the reviewer's concern regarding the reliance on peptide immunization and the generalizability of our findings, we performed additional experiments using full-length protein antigens. We show that immunization with ovalbumin (OVA) and myelin oligodendrocyte glycoprotein (MOG) also induces TCR-like antibodies that specifically recognize the corresponding peptide–MHC-II complexes (**new Supplementary Figures 2a and 2b**). These results demonstrate that the induction of TCR-like antibodies is not restricted to HEL or PLP antigens and can occur following broad protein antigen immunization.

Together, these findings support the physiological relevance of TCR-like antibody generation and suggest that such antibodies can arise in response to commonly used protein antigens, reinforcing both the novelty and broader relevance of the phenomenon.

These data are now described in the Results (page 4-5, lines 62–68) and Discussion (page 15, lines 314–322).

Reviewer's comment 3.

Although the authors present data showing that "TCR-like" antibodies block the binding of a previously described antibody that recognized the specific peptide/MHC-II complex that they are studying, it could be argued that they never directly show that the antibody prevents T cell activation.

Response to this comment:

We appreciate the reviewer's concern and agree that direct functional evidence of T cell inhibition is essential. In addition to demonstrating that TCR-like antibodies block the binding of a previously described reference antibody to the peptide-MHC-II complex, we directly assessed their effects on T cell activation in both *in vitro* and *in vivo* assays.

Specifically, we showed that TCR-like antibodies inhibit activation of antigen-specific TCR reporter cells (**Figures 3c and 7b**) and significantly reduce IL-2 production by antigen-stimulated T cells (**Figure 3d**), which is a primary functional readout of T cell activation *in vitro*. Furthermore, we demonstrated that TCR-like antibodies suppress CD4⁺ T cell–dependent inflammatory responses *in vivo*, including delayed-type hypersensitivity (DTH) and experimental autoimmune encephalomyelitis (EAE) (**Figures 3e, 7d, and 7e**).

Together, these functional data demonstrate that binding of TCR-like antibodies to peptide–MHC-II complexes result in a direct and measurable inhibition of T cell activation and effector function.

Reviewer’s comment 4.

It is not clear whether all appropriate experimental controls have been performed. In particular, did the authors search for "TCR-like" antibodies in non-immunized animals. These could be present if they are germline antibodies (this would also explain their disappearance as the antibody response matures)

Response to this comment:

We thank the reviewer for raising this important point regarding experimental controls. We specifically examined the presence of TCR-like antibodies in non-immunized animals. In all experiments, we detected no TCR-like antibodies in the serum of non-immunized mice, nor in mice immunized with control proteins or control peptides (**Figure 1g and new Supplementary Figures 2a and 2b**).

These findings indicate that TCR-like antibodies are not present as germline-encoded natural antibodies. Instead, they are induced only following specific antigen immunization. Consistent with this interpretation, we also demonstrate that the generation of TCR-like antibodies is strictly CD4⁺ T cell–dependent (as discussed in our response to Comment 1), further supporting the conclusion that these antibodies are generated *de novo* as part of an antigen-driven adaptive immune response.

These data are now described in the Results section (page 5, lines 65-66, 85–88).

Reviewer’s comment 5.

When an antigen is naturally processed and presented via the MHC class II presentation pathway, it is clear from peptidomics data sets that the resulting "epitopes" exist as nested sets of peptides with varying numbers of n-terminal and c-terminal flanking residues. Thus, it would be helpful if the authors could present some evidence that their peptide sequences of interest are naturally processed and presented. In addition, there is some complexity with respect to which HLA/peptide complexes will be present on the surface of antigen presenting cells that have taken up and presented the antigen of interest and which of these would be bound by "TCR-like" antibodies.

Response to this comment:

We agree that demonstrating natural processing and presentation of the target epitope is important for establishing physiological relevance, particularly given that MHC class II peptidomics typically reveals nested peptide sets with variable N- and C-terminal flanking residues. To address this, we performed LC–MS/MS analysis of peptides eluted from MHC-II immunoprecipitated from antigen-presenting cells pulsed with the relevant protein antigen. In these samples, we detected naturally processed HEL-derived peptides that include the C-terminal flanking residues required for induction/recognition of the peptide–MHC-II complex targeted by our TCR-like antibodies (**new Supplementary Figure 2d, Supplementary table 1**). Consistent with this, a previous study reported that MHC-II presentation of the HEL epitope frequently includes peptides containing C-terminal flanking residues (Reference 17).

These results support the conclusion that the specific peptide–MHC-II complex recognized by our TCR-like antibodies is generated during physiological antigen processing and is therefore available for antibody binding *in vivo*. We also agree with the reviewer that multiple related peptide–MHC-II complexes can be displayed simultaneously on antigen-presenting cells. Our findings indicate that, within this naturally occurring nested set, the flanking-residue–containing species is present and thus provides a physiologically relevant target for TCR-like antibody recognition.

This new dataset has been incorporated into the Results section (page 6, line 95-100).

Reviewer's comment 6.

Beyond this, assuming that "TCR-like" antibodies do bind to a given antigen presenting cell, would the most potent biologic effect be to block T cell activation or would this trigger antibody-dependent cellular cytotoxicity and death of the antigen presenting cell? The authors should probably do at least some work to verify this mechanism.

Response to this comment:

We agree that clarifying the mechanism of action—steric blockade of T cell recognition versus Fc-mediated effector functions such as ADCC—is important for interpreting the biological effects of TCR-like antibodies. To address this, we performed both *in vitro* and *in vivo* assays to evaluate ADCC.

In vitro, we observed antibody-dependent cytotoxicity that was dependent on both the concentration of the TCR-like antibody and the presence of the cognate HEL peptide presented by MHC-II on target cells (**new Supplementary Figure 5e**). *In vivo*,

administration of the TCR-like antibody reduced HEL-specific B cells, which are expected to present the relevant pMHC-II complexes at high density (**new Supplementary Figures 5f and 5g**). Together, these results support the conclusion that TCR-like antibodies can mediate Fc-dependent effector activity in addition to blocking peptide–MHC-II recognition; however, it remains unclear which mechanism is predominant *in vivo*.

These new data are described in the Results (page 9, lines 179–189) and discussed in the Discussion (page 14, lines 306–309).

Reviewer’s comment 7.

It could be argued that the degree of clinical improvement in the EAE model is not that impressive. Do these "TCR-like" antibodies really provide a promising therapy for autoimmune disease?

Response to this comment:

We agree with the reviewer that the magnitude of clinical improvement observed in the therapeutic EAE setting is modest, although it reaches statistical significance. We therefore view these results as proof-of-concept that endogenously induced or administered TCR-like antibodies can modulate antigen-specific autoimmune pathology, rather than as evidence that the current approach is already optimized for therapeutic efficacy.

Consistent with this, the TCR-like antibody clones analyzed in this study were not selected through extensive affinity maturation/engineering workflows typically used for drug development, and the kinetics of antibody induction *in vivo* may also limit observable effects in stringent therapeutic models. Taken together, our findings support the feasibility of antigen-specific immune modulation via TCR-like antibodies while underscoring that further optimization (e.g., selection of higher-affinity clones and/or refinement of immunization conditions) would be required for translational development.

These points are discussed in the Discussion section (page 16, lines 344–350).

Reviewer’s comment 8.

The discussion should include more critical thinking about the experimental results and potential limitations of the experimental system used. Most notably, given the absence of physiologic immunization and the decline in the levels of "TCR-like" antibodies by two

weeks after immunization, it is not clear how much a part of the "natural" immune response these are. Thinking about these antibodies as a potential therapeutic, it seems likely that additional refinement of the antibody sequence to improve its affinity would be needed to create a monoclonal that could be put forward as a drug candidate. Could this be done without compromising their specificity?

Response to this comment:

We thank the reviewer for encouraging a more critical discussion of the experimental system and the implications/limitations of our findings.

Regarding whether these antibodies are part of a “natural” immune response, we agree that our immunization conditions do not fully recapitulate physiological antigen exposure (e.g., infection or vaccination) and that TCR-like antibody levels decline within ~2 weeks after immunization. We have therefore expanded the Discussion to more explicitly acknowledge these limitations and to clarify interpretation of the kinetics. Importantly, our data indicate that TCR-like antibodies are not pre-existing germline-encoded natural antibodies, but rather are induced *de novo* during antigen-specific adaptive immune responses: they are IgG, are not detectable in non-immunized mice, and their production is strictly CD4⁺ T cell–dependent (see responses to comments 1 and 4). The transient nature of their appearance may limit sustained suppressive activity, but may also be consistent with a time-restricted regulatory role that attenuates excessive immune activation without broadly suppressing protective immunity.

These points have been incorporated into the Discussion (page 16-17, lines 363-366).

Regarding therapeutic potential, we agree that the TCR-like antibody clones analyzed in this study may not represent the highest-affinity antibodies and that additional optimization would likely be required for translational development. In therapeutic antibody discovery, improved monoclonal candidates are typically obtained through repeated antigen exposure and/or *in vitro* engineering with stringent screening to enrich for maximal affinity maturation. Because TCR-like antibody induction is CD4⁺ T cell-dependent, affinity maturation is expected to occur; moreover, affinity optimization can be pursued while preserving specificity by employing standard selection strategies that include rigorous counter-selection against MHC alone, peptide alone, and irrelevant peptide-MHC complexes, alongside positive selection for the cognate peptide-MHC-II complex.

These points have been incorporated into the Discussion (page 16, lines 344-350).

Reviewer #2:

General comment

This study reports the novel discovery that immunization with protein antigens, such as HEL or peptides containing flanking residues (FRs) can induce the production of TCR-like antibodies (iTabs) in mice. These iTabs specifically recognize peptide-MHC class II complexes and block TCR engagement, offering a potential strategy for suppressing antigen-specific CD4⁺ T cell responses. The authors determined a cryo-EM structure of the iTab Fab in complex with the HEL-MHC II complex, revealing the detailed iTab recognition of the peptide, the C-terminal FR of the peptide, and the MHC II molecule. Overall, the findings are innovative, well-supported by experimental evidence, and provide a promising new approach for controlling autoimmune diseases.

Response to this comment:

We thank the reviewer for their positive and constructive assessment of our manuscript and for recognizing the novelty and potential significance of our findings. We appreciate the reviewer's evaluation of both the immunological evidence supporting antigen-induced TCR-like antibodies (iTabs) and the structural insights provided by the cryo-EM analysis of the iTab-peptide-MHC-II complex. We have incorporated the reviewer's additional comments and suggestions in the revised manuscript as detailed below.

Reviewer's comment 1

Lines 89-90, "To analyze the role of C-terminal and N-terminal FR in inducing iTabs, we generated recombinant HEL or Sm-P40 peptides containing C-terminal WWA FR or N-terminal PKS FR." The authors may want to explain the rationale for choosing the WWA and PKS residues to elicit the antibody response.

Response to this comment:

Based on our results showing that HEL requires the C-terminal WWA and Sm-P40 requires the N-terminal PKS for iTab induction, we generated FR-swapped chimeric peptides (PKS-HEL and Sm-P40-WWA) to determine whether iTab induction is driven primarily by the flanking residues or depends on a specific FR-core pairing.

We revised the description on this point in the Results section (page 7, lines 121-124, 130-133).

Reviewer's comment 2

Can the authors report the buried surface area values of the antibody interface with the peptide (excluding the C-terminal WW), the C-terminal WW residues, and the MHC-II molecule?

Response to this comment:

We thank the reviewer for this request. We have now calculated buried surface area (BSA) values to quantify the AiTab Fab interface with each component of the pMHC-II complex. Specifically, the Fab buries 291.0 Å² on the HEL peptide core (excluding the C-terminal WW), 301.4 Å² on the C-terminal WW residues, and 690.3 Å² on the MHC-II molecule (**new Supplementary Table 3**). Notably, the C-terminal WW residues account for approximately 51% of the total Fab-peptide buried surface area.

These values have been added to the Results section (page 11, lines 221–227).

Reviewer's comment 3

Line 371 “The construction of soluble MHC class II molecules has been previously described.” Please provide the appropriate references here.

Response to this comment:

We thank the reviewer for this suggestion. We have now added the appropriate reference to the relevant statement in the main text (page 25, line 583).

Reviewer's comment 4

Fig 5b legend, the PDB code 1IAK should be changed to 1IAK.

Response to this comment:

We apologize for this typographical error. We have corrected the PDB code to 1IAK in the Fig. 5b legend.

Reviewer's comment 5

It would be helpful if the authors could change the antibody numbering to a standardized numbering scheme, such as IMGT, Kabat, or Chothia.

Response to this comment:

We have re-numbered the antibody residues using the IMGT numbering scheme and have provided the updated residue mapping in the **new Supplementary Table 4**.

Reviewer's comment 6

Fig 5f: The current color scheme uses pink and magenta, which are difficult to distinguish. The authors may want to change the colors to improve visual clarity. In addition, please include the chain IDs for K63 and Y60. In the figure, E59 appears to be an alanine.

Response to this comment:

We thank the reviewer for these helpful suggestions and have addressed each point in **Fig. 5f**. We changed the color of CDR2 from pink to wheat to improve visual clarity, and we added the chain IDs for K63 (β chain) and Y60 (β chain). Regarding MHC β E59, the side chain was modeled as alanine due to low local map quality in this region. We have updated the Fig. 5f legend to clarify this point (e.g., “The side chain of MHC β E59 is omitted because it was modeled as alanine.”).

Reviewer #3:

General comment

The central question of the study was whether peptide:MHC-II antibodies are generated during the immune response, and if these antibodies serve as negative immune regulators to limit T cell effector function. The authors also asked whether longer peptides were more effective at generating p:MHC-II antibodies compared to short peptides lacking MHC-II flanking residues.

The central question has broad appeal to the immunology and medical community yet the rationale behind this question is not well supported by observations in humans or mice. While the authors provide compelling data that p:MHC-II antibodies are generated following immunization with HEL and PLP-derived peptides, the data does not robustly support the authors claims that these antibodies serve as a negative regulator of T cell immunity. Furthermore, the authors claim that pMHC-II antibodies generated during peptide immunization could potentially be used to control autoimmune diseases but the authors do not specifically discuss how this may be applied to human health. Many studies have used pMHC-II antibodies as biologics in preclinical models of human disease so that is not novel. How do the authors envision peptide immunization to prevent or treat autoimmune disease?

Response to General comment:

We thank the reviewer for their thoughtful and constructive comments regarding the novelty, mechanistic interpretation, and translational relevance of our study.

1) Novelty and physiological relevance of TCR-like antibodies

As the reviewer noted, TCR-like (pMHC-specific) antibodies have been described previously, primarily through immunization with recombinant peptide–MHC complexes. In contrast, to our knowledge, there have been no prior reports demonstrating the endogenous generation of TCR-like antibodies during an *in vivo*, antigen-specific immune response elicited by immunization with whole antigens or peptides alone. Our study provides evidence that such antibodies can arise naturally *in vivo* in a T cell–dependent manner following antigen exposure. This observation suggests that TCR-like antibodies may represent a previously underappreciated component of adaptive immune responses. Moreover, because these antibodies are generated concomitantly with antigen-specific T cell responses, it is plausible that they may contribute to immune regulation by attenuating excessive T cell activation, which is known to be harmful under certain conditions.

These points have been incorporated into the Discussion (page 14, lines 301–312).

2) Conceptual advance and relevance to human disease

The core novelty of our work lies in the concept of inducing TCR-like antibodies endogenously through antigen immunization, rather than administering exogenous pMHC-II antibodies as biologics. We propose that this approach could, in principle, provide more durable and antigen-specific immune modulation compared to passive antibody transfer. Although translation to humans was not directly addressed in this study, owing to the complexity of HLA polymorphism and the limited availability of well-defined human epitopes, our findings establish a conceptual framework that may inform future strategies for antigen-specific immune regulation in autoimmune or allergic diseases.

These points have been incorporated into the Discussion (page 17, lines 373–376).

Reviewer’s comment 1

Immunizing mice with mutant peptide, that cannot activate T cells, prior to wild type peptide immunization, leads to a reduction in T cell proliferation and cytokine production (Fig 4d) and EAE disease (Fig 7f). The authors claim that reduced T cell proliferation and effector function, and EAE disease, is a direct result of endogenous production of pMHC-II antibodies. As an alternative explanation, could the initial peptide immunization generate anti-peptide (not pMHC-II) specific antibodies that then neutralize the wild type peptide, preventing it from being taken up by APCs and being efficiently presented to T

cells? Can the authors show by ELISA whether HEL or PLP peptide specific antibodies are generated by immunization? For the experiment described in figure 7c, can serum from immunized mice be used to suppress T cell responses instead of purified antibody? If so, perhaps the serum could be used to determine if reduced T cell responses result from pMHC-II antibodies vs peptide antibodies. I wonder if the immune serum could be absorbed using peptide prior to transfer to eliminate the peptide specific antibodies, leaving the pMHC-II antibodies in the serum?

Response to this comment:

We thank the reviewer for raising this important alternative explanation, namely that anti-peptide antibodies generated during the initial immunization might neutralize the wild-type peptide and thereby reduce antigen presentation and T cell activation. To address this possibility, we performed the following additional experiments.

First, we measured anti-PLP peptide antibody titers by ELISA and found that immunization with the mutant peptide, either with or without FR, induced comparable levels of anti-peptide antibodies (**new Supplemental Figure 7a**). Despite similar anti-peptide antibody titers, only immunization with the FR-containing peptide resulted in suppression of T cell responses, indicating that anti-peptide antibodies alone are unlikely to account for the observed suppressive effect.

Second, we examined whether immune serum IgG could suppress antigen-specific T cell responses. As shown in **Figure 3c**, polyclonal IgG purified from serum of mice immunized with FR⁺ peptide suppressed antigen-specific T cell responses, whereas serum from mice immunized with FR⁻ peptide—despite containing anti-peptide antibody—did not. These findings further argue against a major role for anti-peptide antibodies in mediating T cell suppression.

These points are described in the Results section (page 13, lines 268–270 and page 14, 293–296, and page 8, 158-162).

Reviewer's comment 2

Supplemental figure 7 shows a model whereby pMHC-II antibodies are involved in the contraction phase of the immune response. Can the authors show antigen specific T cell expansion kinetics in WT vs BCR transgenic mice to support the idea that antibodies are helping to resolve the inflammatory process, creating a steeper or earlier drop in T cell number following expansion.

Response to this comment:

We thank the reviewer for this insightful suggestion. Analyzing antigen-specific T cell expansion and contraction kinetics in BCR-transgenic mice would be informative for understanding how antibodies influence immune resolution. However, we believe that this experimental approach is not well suited to addressing the central question of the present study.

In BCR-transgenic mice, antibody production is constitutive and not regulated by antigen-specific immune activation. Consequently, the transgene-encoded antibodies are present independently of antigen exposure, unlike in our immunization model, in which TCR-like antibodies are induced during an antigen-specific immune response. Because the primary novelty of our study lies in demonstrating the endogenous, antigen-dependent induction of TCR-like antibodies, kinetic analyses using BCR-transgenic mice would not accurately reflect the physiological context we aimed to investigate.

We therefore consider that such experiments, while valuable for future mechanistic studies, fall beyond the scope of the current work.

Reviewer's comment 3

The authors use two model antigens, HEL and PLP, to draw broad conclusions about the need for flanking residues in pMHC-II antibody generation. Structural analysis of pMHC antibodies has shown that these antibodies can have several modes of binding antigen, including TCR-like footprint across the pMHC (PMID: 36593402). To make the claim that pMHC antibodies are generally directed against flanking residues I suggest the authors include a wider range of peptides, especially peptides with bulky or charged residues in the MHC groove but exposed for TCR binding, like 2W or 3K.

Response to this comment:

We thank the reviewer for highlighting the structural diversity of pMHC antibody binding modes and for suggesting the inclusion of peptides containing bulky or charged residues exposed for TCR recognition. In response to this suggestion, we analyzed additional HEL-derived peptides harboring mutations in TCR-facing residues within the MHC-binding groove. Specifically, we introduced 2W substitutions at positions 53/54 and 59/60, and 2K substitutions at positions 59/60.

Immunization with these mutated peptides failed to induce antibodies recognizing the WT peptide–MHC-II complex (**new Figure 2b and Supplemental Figure 3b**). This result

indicates that introducing bulky/charged substitutions at TCR-exposed positions is not sufficient to generate antibodies that recognize the WT peptide–MHC-II complex, consistent with a critical contribution of flanking residues to the induction/recognition described here. Accordingly, these additional data further support our conclusion that flanking residues play a critical role in the induction of the pMHC-II antibodies described in this study.

These new data have been incorporated into the Results section (page 6-7, lines 112–119).

Reviewer’s comment 4

Little information was provided about the frequency of B cells that are pMHC-II-specific and whether they underwent T dependent affinity maturation prior to generating hybridomas. What fraction of hybridomas screened were pMHC-II specific? The significance of the work would be enhanced by a more thorough description of the frequency of pMHC-II-specific B cells before and after immunization, how this changes depending on the peptide epitope, and strain of mice. The translational impact would also benefit by determining whether pMHC-II antibodies naturally exist in other autoimmune disease models, such as anti-InsB:IAg7 in NOD mice (PMID: 24550292) or anti-citFib:DR4 in DR4-IE mice (PMID: 18391064). Furthermore, the authors could use pMHC tetramers to screen human PBMC for pMHC antibodies against viral antigens induced by vaccination.

Response to this comment:

We thank the reviewer for these insightful suggestions regarding the frequency, maturation status, and broader relevance of pMHC-II-specific B cells.

T cell dependence and affinity maturation

We demonstrated that pMHC-II antibody production is CD4 T cell-dependent by immunizing CD4 T cell-depleted mice, in which pMHC-II antibodies were not detected (**new Supplemental Figure 2c**). In addition, pMHC-II antibodies were undetectable in non-immunized mice. Sequence analysis of 11-72 immunoglobulin genes from pMHC-II-specific hybridomas revealed multiple somatic mutations in both heavy (G36A, S57R) and light (A32T, S114R) chains, consistent with T cell-dependent affinity maturation. The new data and interpretations are now described in the Results (page 5, lines 81–88) and Discussion (page 15, lines 332–336).

Frequency of pMHC-II-specific B cells and hybridomas

Despite these findings, unfortunately, we were unable to directly detect pMHC-II-specific B cells using recombinant pMHC-II proteins from mice immunized with HEL protein, suggesting that their frequency is low. Consistent with this, hybridomas generated from whole draining lymph nodes of immunized mice yielded pMHC-II-specific clones at a frequency of less than 1% of total screened hybridomas.

These points were described in the Results sections (page 8-9, lines 163-165) and Method sections (page 21, lines 477-478).

Other autoimmune models

In response to the reviewer's suggestion, we examined the presence of anti-InsB:I-A^{g7} pMHC-II antibodies in cyclophosphamide-treated NOD mice but did not detect such antibodies (**new Supplemental Figure 7d**). While this result does not allow definitive conclusions, it suggests that iTab production may not be a universal feature across autoimmune models and may depend on antigenic context or immune environment. Notably, insufficient induction of iTabs could plausibly contribute to dysregulated autoreactivity in some settings, a possibility that warrants further investigation.

The results has been added to the Results sections (page 13, lines 273-276) and the Discussion sections (page 17, line 366-371)

Generality and translational considerations

We agree that assessing pMHC-II antibodies in additional disease models and in humans would be highly informative. However, given the extensive diversity of HLA class II alleles, target antigens, and the transient nature of pMHC-II antibody production observed in our study, such analyses would require large-scale cohort studies and are beyond the scope of the current work.

Instead, to broaden the relevance of our findings beyond HEL and PLP, we analyzed immune responses to OVA and MOG and found that pMHC-II antibodies were also induced during antigen-specific immune responses to these antigens (**new Supplementary Figures 2a and 2b**). These additional data support the generalizability of our observations across multiple antigen systems.

These points were described in the Results (page 4-5, lines 62–68) and Discussion (page 16, 344-350)

Minor points:

1) A major conclusion is based on the experiment shown in Fig. 7f yet the results are not very robust, with a modest, yet significant, reduction in peak disease. It states that this experiment was representative of 3. I suggest the authors show all 3 experiments in the supplement so help support their conclusion that pMHC-II antibodies can impact autoimmune disease.

Response to this comment:

We thank the reviewer for this suggestion. All three independent EAE experiments corresponding to Fig. 7f are now included in the Supplementary Information (**Supplementary Figure 7f**), providing additional support for the reproducibility of the observed reduction in disease severity.

2) The introduction included results and a conclusion statement rather than a thorough explanation of the rationale for conducting the research. Overall, the writing would benefit from further editing for clarity and formatting.

Response to this comment:

We have revised the Introduction to remove result-oriented statements and to more clearly articulate the rationale and background for the study. The revised Introduction has also been edited for clarity and formatting in accordance with the reviewer's suggestions. (page 3, lines 34-38)

Point by point response to the Reviewer's comments.

Reviewer #1:

The additional experimental data and revisions are sufficient to allay my concerns.

Response:

We sincerely thank the reviewer for evaluating our manuscript and for the constructive comments, which have greatly improved the quality of our paper. We are very glad that our additional experimental data and revisions have fully addressed your concerns.

Reviewer #2:

The authors have addressed all my concerns.

Response:

We sincerely thank the reviewer for evaluating our manuscript and for the constructive comments, which have greatly improved the quality of our paper. We are very glad that our additional experimental data and revisions have fully addressed your concerns.

Reviewer #3:

The authors have sufficiently addressed all of my concerns. I recommend the manuscript for publication.

Response:

We sincerely thank the reviewer for evaluating our manuscript and for the constructive comments, which have greatly improved the quality of our paper. We are very glad that our additional experimental data and revisions have fully addressed your concerns.